# From Invariant Representations to Invariant Data: Provable Robustness to Spurious Correlations via Noisy Counterfactual Matching

## Abstract

Models that learn spurious correlations from training data often fail when deployed in new environments. While many methods aim to learn invariant representations to address this, they often underperform standard empirical risk minimization (ERM). We propose a data-centric alternative that shifts the focus from learning invariant representations to leveraging invariant data pairs—pairs of samples that should have the same prediction. We prove that certain counterfactuals naturally satisfy this invariance property. Based on this, we introduce Noisy Counterfactual Matching (NCM), a simple constraint-based method that improves robustness by leveraging even a small number of *noisy* counterfactual pairs—improving upon prior works that do not explicitly consider noise. For linear causal models, we prove that NCM's test-domain error is bounded by its in-domain error plus a term dependent on the counterfactuals' quality and diversity. Experiments on synthetic data validate our theory, and we demonstrate NCM's effectiveness on real-world datasets.

## 1 Introduction

Spurious correlations are misleading patterns in the training data. The relationships between features and the target do not hold across domains or environments. Models trained on such correlations may perform well on their training distribution yet fail to generalize once the environment changes, because the correlations reflect coincidental or confounded associations rather than true causal links. Addressing spurious correlations is therefore critical for building models that remain reliable under distribution shift—especially in high-stakes domains such as healthcare, finance, and public services. Spurious correlation falls under the broader problem of domain generalization (DG), which seeks to generalize to new unseen test domains beyond the original training domains.

Invariant representation learning tackles the DG problem by forcing some distributional property of the representation to be stable across domains (Peters et al., 2016; Li et al., 2018b;a; Arjovsky et al., 2019). Approaches range from matching the marginal $p(h(\mathbf{x}))$ or conditional $p(h(\mathbf{x})|\mathbf{y})$ (Li et al., 2018b), to causality-inspired objectives such as Invariant Causal Prediction (ICP) (Peters et al., 2016) and Invariant Risk Minimization (IRM) (Arjovsky et al., 2019), which match the conditional $p(\mathbf{y}|h(\mathbf{x}))$. Both MatchDG (Mahajan et al., 2021) and Domain Invariant Representation Learning with Domain Transformations (DIRT) (Nguyen et al., 2021) consider a two stage approach to invariant representation learning. In the first stage, they estimate a mapping between domain distributions; MatchDG uses iterative contrastive learning to find data pairings between domains while DIRT used StarGAN Choi et al. (2018) to learn an explicit map between domains. In the second stage, they add an invariant representation regularization term that encourages the latent representations of pairs to be close. Although theoretically grounded, these methods often underperform empirical risk minimization (ERM) on modern benchmarks (Gulrajani and Lopez-Paz, 2021; Koh et al., 2021; Bai et al., 2024), which may be due to the strong assumptions that do not necessarily hold in practice.

Inspired by the recent DG works that leverage additional data beyond the standard DG setup Blanchard et al. (2011b), we consider the following research question: **Can shifting the focus from learning invariant *representations* to leveraging invariant *data* provide a more direct and practical path toward domain generalization?** Figure 1 illustrates the core intuition of why invariant

pairs could be useful for robustness. This can be viewed as a *data-centric viewpoint* of MatchDG and DIRT that focuses on estimating a robust classifier *given* data pairs between domains (corresponding to stage two of MatchDG and DIRT). While at first glance it may seem that collecting such invariant pairs would be infeasible, we suggest that they could be reasonably acquired in practice under certain scenarios. For example, when the spurious correlations are artifacts of a measurement process (e.g., x-ray machine, microscope, staining methodology, etc.), then an invariant pair could be collected by measuring the same specimen under two different environments (e.g., send the same patient to two x-ray machines). Second, when a domain expert can identify spurious features, they can directly edit spurious features of the sample. An example of this would be using image editing software including AI-based image editing to change the background of an image while keeping the subject the same (e.g., putting a cow on a boat or a fish in a desert).

While these are some natural ways to collect such pairs, the focus of our paper is to theoretically and empirically analyze whether invariant pair data *could* be helpful for robustness—i.e., this is a future-looking paper. Indeed, the current lack of invariant pair datasets should not lead to the conclusion that invariant pairs could not be collected, but rather that the utility of such pairs is not obvious. We hypothesize that invariant data pairs could enable a data-driven way to *implicitly* encode knowledge of spurious correlations instead of requiring explicit specification (e.g., specifying a causal graphical model). To explain via an analogy, collecting invariant pairs could be to spurious correlations as collecting class labels is to classification. In both cases, explicitly defining the target object (either spurious correlations or class) can be very challenging if not impossible but implicitly defining them through data is significantly easier.

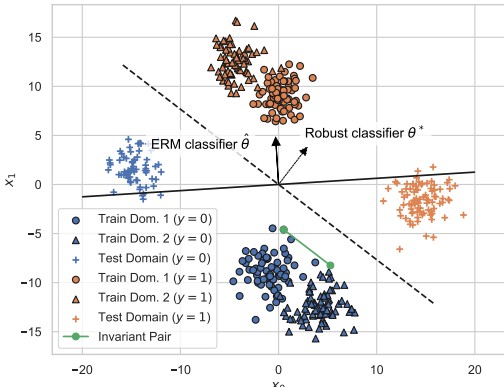

Figure 1: While ERM $\hat{\theta}$ on the training domains (circles and triangles) is not robust to the change in spurious feature in the unseen test domain (pluses), a robust linear classifier $\theta^*$ can be estimated by making the classifier orthogonal to the difference between a *single* invariant pair (green line). The color represents label y.

The natural next question is: How can these invariant pairs be used for robustness? MatchDG (Mahajan et al., 2021) and DIRT (Nguyen et al., 2021) propose a simple regularization that encourages the latent representations of pairs to be close. However, these works do not address two questions critical for a data-centric viewpoint: *How do we theoretically and practically handle* **noise** *in the invariant data pairs?* **How many pairs** *are needed for robustness theoretically and empirically?* The first addresses the inevitable noise incurred when collecting or creating invariant data pairs, i.e., they are not perfect pairs. The second addresses the practical question of whether it may be cost-effective to collect such pairs, which may be costly to obtain (though not necessarily).

To address these data-centric questions, we analyze the theoretical guarantees and trade-offs of using **invariant pairs** (pairs of inputs that should have the same prediction), focusing on the linear setting. We formalize the spurious correlation setting using a causal perspective, proving that certain spurious counterfactuals naturally create these invariant pairs. Based on this, we introduce **Noisy Counterfactual Matching (NCM)**, a simple method that adds a linear constraint to ERM. This constraint, derived from the SVD of the differences between pairs, forces the model to ignore the spurious features identified by the pairs. We prove that NCM is robust to spurious correlations even with a small number of noisy pairs, and we validate our findings empirically. We summarize our main contributions as:

- We introduce Noisy Counterfactual Matching, a simple, data-centric method that adds a constraint to ERM to improve robustness to spurious correlation using a small set of noisy invariant pairs.

- We theoretically analyze NCM's robustness by proving an out-of-domain error bound that decomposes into the in-domain risk and a term dependent on the quality of provided pairs.

- We show that the number of pairs needed can scale linearly with the spurious feature dimension.

- We empirically validate our theory, demonstrating improved robustness on synthetic data and on real-world benchmarks via linear probing on a pretrained CLIP model.

**Notation:** We use lowercase letters to denote random variable (e.g., y), bold lowercase letters to denote random variables (e.g., $\mathbf{x}$), and bold italic letters for their realizations (e.g., $\boldsymbol{x}$). Capital letters represent matrices or constants (e.g., $U$, $N$), while calligraphic letters denote sets, domains, or ranges (e.g., $\mathcal{M}$, $\mathcal{E}$, $\mathcal{X}$). The notation $[N]$ represents the index set $1, 2, \ldots, N$. We denote the $r$-th largest singular value by $\sigma_r$, and $Q_r$ denotes the $r$-largest singular values' corresponding singular vectors. $\|A\|_\Lambda := \|A^\top \Lambda^{1/2}\|$ is Mahalanobis-induced spectral norm.

## 2 PROBLEM SETUP

To formalize the goal of robustness to spurious correlations, we consider a set of domains $\mathcal{E}$ where their difference are on spurious features. Our goal is to find the optimal classifier on these domains.

**Definition 1** (Optimally Robust Classifier). *Given a set of environments $\mathcal{E}$, the optimally robust classifier is defined as:*

$$h_\mathcal{E}^* := \arg\min_h \max_{e \in \mathcal{E}} \mathbb{E}_{(\mathbf{x}, \mathbf{y}) \sim \mathbb{P}_e}[\ell(h(\mathbf{x}), \mathbf{y})],$$

*where the optimization is over all possible predictive functions $h$.*

In this work, we consider a setting where, in addition to the training set, a small dataset of *invariant data pairs* is available. While our objective remains the same as the standard domain generalization (DG) setup (Blanchard et al., 2011a), which is to achieve strong performance on an unseen test domain, our data requirements differ. In contrast with DG's requirement of labeled training sets from multiple environments, our method requires labeled training data from one domain and a small group of invariant data pairs as defined below.

**Definition 2** (Invariant Pair). *Given a set of environments $\mathcal{E}$, a pair of distinct inputs $(\boldsymbol{x}, \boldsymbol{x}')$ with $\boldsymbol{x} \neq \boldsymbol{x}'$ is an invariant data pair if and only if the predictions under the optimally robust classifier are equal, i.e., $h_\mathcal{E}^*(\boldsymbol{x}) = h_\mathcal{E}^*(\boldsymbol{x}')$, where $h_\mathcal{E}^*$ is defined as in Definition 1.*

Intuitively, invariant data pairs are inputs that should have the same prediction under a robust model. For example, in medical diagnosis, X-rays from two different machines of the same patient should yield the same probabilities, without requiring knowledge of the patient's actual diagnosis.

### 2.1 CAUSALITY PRELIMINARIES

To formally define the set of spurious correlation environments and their corresponding invariant pairs, we introduce some related concepts in causality. In summary, we consider that each domain (or environment[1]) corresponds to a distinct structural causal model (SCM) (Pearl, 2009, Definition 7.1.1), the differences of the SCMs are equivalent to interventions, and counterfactuals are based on applying two different SCMs to the same exogenous noise. First, we formally define an SCM.

**Definition 3** (Structural Causal Model (Pearl, 2009, Definition 7.1.1)). *An SCM $\mathcal{M}$ is represented by a 3-tuple $\langle \mathcal{U}, \mathcal{V}, \mathcal{F} \rangle$, where $\mathcal{U}$ is the set of exogenous noise variables, $\mathcal{V}$ is a set of causal variables, and $\mathcal{F} := \{f_1, f_2, \ldots, f_m\}$ denotes the set of causal mechanisms for each causal variable in $\mathcal{Z}$ given its corresponding exogenous noise and parents, i.e., $\mathbf{v}_i = f_i(\mathbf{u}_i, \mathbf{v}_{\mathrm{Pa}(i)})$.*

We denote causal mechanism in domain $e$ as $\mathcal{F}_e := \{f_{e,1}, f_{e,2}, \ldots, f_{e,m}\}$. Given this, we consider two notions when comparing two different causal models: intervention set and counterfactuals.

**Definition 4** (Intervention Set). *Given two SCMs $\mathcal{M}$ and $\mathcal{M}'$ defined on the same set of exogenous noise and causal variables, the intervention set is defined only in terms of their causal mechanisms $\mathcal{F}$ and $\mathcal{F}'$ respectively $\mathcal{I}(\mathcal{F}_e, \mathcal{F}_{e'}) = \{i : f_{e,i} \neq f_{e',i}\}$.*

Note that this definition allows multiple types of intervention including soft, hard or do-style interventions. We now define counterfactual pairs as applying two SCMs to the same exogenous noise based on the original definition of counterfactuals in SCMs (Pearl, 2009, Definition 7.1.5).

**Definition 5** (Counterfactual Pair). *A pair of causal variable realizations $(\boldsymbol{v}_\mathcal{A}, \boldsymbol{v}'_\mathcal{A})$ where $\mathcal{A} \subseteq \mathcal{V}$ is a subset of causal variables is a counterfactual pair between two SCMs $\mathcal{M}$ and $\mathcal{M}'$ (with the same set of exogenous noise variables and causal variables) if and only if there exists a exogenous noise realization $\boldsymbol{u}$ such that $\boldsymbol{v}_\mathcal{A}$ is the solution to $\mathcal{M}$ and $\boldsymbol{v}'_\mathcal{A}$ is the solution to $\mathcal{M}'$.*

---

[1]We will use domain and environment interchangeably.

Note that this is different than *estimating* counterfactuals given some factual evidence, which would require the three steps of abduction, action, and prediction. Rather, here we simply define the theoretic notion of a CF pair between two SCMs. However, in practice, we expect that perfect CF pairs will not be feasible so we focus on providing theoretic analysis of noisy CF pairs.

## 2.2 Latent Spurious Correlations

After introducing the causal preliminaries, we now formalize the latent spurious correlations by specifying the collection of SCMs that define domains. This follows many latent SCM multi-domain works (Liu et al., 2022; Zhang et al., 2023; von Kügelgen et al., 2023; Zhou et al., 2024).

**Definition 6** (Class of Latent Domain SCMs). *Letting $\mathcal{E}$ denote the set of domains, a latent domain SCM class is a set of latent SCMs $\mathcal{M}_{\mathcal{E}} = \{\mathcal{M}_e\}_{e \in \mathcal{E}}$ such that:*

1. *The causal models share the same set of exogenous noise variables, causal variables, and exogenous noise distribution $\mathbb{P}_{\mathcal{U}}$.*
2. *The causal variables $\mathcal{V}$ are split into observed variables $\mathcal{X} \cup \mathcal{Y}$ and latent variables $\mathcal{Z}$.*
3. *The models share the same causal mechanisms for the observed variables, denoted by $g_{\mathbf{x}}$ and $g_{\mathbf{y}}$, and can only have latent variables in $\mathcal{Z}$ as parents.*

*The latent causal mechanisms for the $i$-th variable in $\mathcal{Z}$ for the $e$-th domain will be denoted as $f_{e,i}$, and the induced distribution over the observed random variables for each domain will be denoted by $\mathbb{P}_e(\mathbf{x}, \mathbf{y})$. The intervention set among the SCM class is defined as $\mathcal{I}(\mathcal{F}_{\mathcal{E}}) \coloneqq \bigcup_{e,e' \in \mathcal{E}} \mathcal{I}(\mathcal{F}_e, \mathcal{F}_{e'})$.*

We now give our primary spurious correlation assumption that the domains in the class can only intervene on spurious latent variables with respect to the target variable y, i.e., non-ancestors of y.

**Assumption 1** (Spurious Correlation Latent SCM Class). *Any variable in the intervention set must be non-ancestors of y, i.e., $\mathcal{I}(\mathcal{F}_{\mathcal{E}}) \cap \mathrm{Anc}(\mathbf{y}) = \emptyset$. Equivalently, all domains must share the mechanisms for ancestors of y, i.e., $f_{e,i} = f_{e',i}, \forall i \in \mathrm{Anc}(\mathbf{y}), e, e' \in \mathcal{E}$.*

This assumption defines the scope of our work to spurious correlation, which limits the types of shift that we could see at test time to only spurious features, i.e., non-ancestors of y. However, this assumption does not limit the strength of these shifts. Intuitively, if the sample x encodes information about a descendant of y, a predictor trained on x cannot be invariant across interventional distributions because x is a collider of all the latent causal variables and thus e and y will not be d-separated. This is true even when interventions only target spurious features. We include an illustration of causal DAG Figure 3 and a detailed explanation in Appendix A.1.

## 3 Handling Noise in Invariant Pairs via Noisy Counterfactual Matching

Our goal in this section is to show how counterfactual pairs can be used to improve robustness. Specifically, we first discuss the relationship between spurious counterfactuals and invariant pairs. Motivated by this, we introduce the noisy counterfactual matching (NCM) method, which aims to identify and recover the spurious subspace by leveraging potentially noisy CF pairs drawn exclusively from the training domains.

**Spurious Counterfactuals are Invariant Pairs.** Given the causal model setup in Section 2, we can now prove that counterfactuals within a spurious correlation latent SCM class are invariant pairs w.r.t. the corresponding domain distributions.

**Proposition 1** (Spurious Counterfactuals are Invariant Pairs). *Given a spurious correlation latent SCM class $\mathcal{M}_{\mathcal{E}}$ and a strictly convex loss function $\ell$, any observed counterfactual pair $(\boldsymbol{x}_e, \boldsymbol{x}_{e'})$ between $\mathcal{M}_e \in \mathcal{M}_{\mathcal{E}}$ and $\mathcal{M}_{e'} \in \mathcal{M}_{\mathcal{E}}$ will be an invariant pair w.r.t. the optimally robust classifier $h_{\mathcal{E}}^*$ based on $\ell$ induced by the domain distributions $\{\mathbb{P}_e\}_{e \in \mathcal{E}}$ almost surely, i.e., $h_{\mathcal{E}}^*(\boldsymbol{x}_e) = h_{\mathcal{E}}^*(\boldsymbol{x}_{e'})$.*

See proof in Appendix C.1. This elegantly connects spurious counterfactuals and invariant pairs (though again we note that invariant pairs could be defined for other perspectives). The natural next question is: *Is it possible to collect such pairs in reality?* We argue that while perfect counterfactual pairs are not possible, noisy or approximate counterfactual pairs could be reasonably simple to collect in certain scenarios (see Appendix A.2)

**Noisy Counterfactual Matching (NCM).** Given a set of CF pairs solely from the training domains $\{(\boldsymbol{x}_{e_j}, \boldsymbol{x}_{e_j \to e'_j})\}_{j=1}^k$, by Proposition 1, those counterfactuals are invariant pairs, i.e., $h_{\mathcal{E}}^*(\boldsymbol{x}_e) = h_{\mathcal{E}}^*(\boldsymbol{x}_{e_j \to e'_j})$. Therefore, it is natural to consider a simple CF pair-matching method that augments empirical risk minimization (ERM) with a constraint enforcing the outputs of each pair to be equal:

$$\min_h \mathbb{E}_{(\mathbf{x},\mathbf{y}) \sim \mathbb{P}_{\text{train}}}[\ell(h(\mathbf{x};\theta), \mathbf{y})], \quad \text{s.t.} \ \ h(\boldsymbol{x}_{e_j}; \theta) = h(\boldsymbol{x}_{e_j \to e'_j}; \theta) \ \ \forall j, \tag{1}$$

where $\ell(\cdot, \cdot)$ measures data fidelity by using the prediction function $h$. If $h$ is linear parameterized by $\theta$, the constraint simplifies to $\theta^\top \delta_j = 0, \forall j$, where $\delta_j := \boldsymbol{x}_{e_j} - \boldsymbol{x}_{e_j \to e'_j}$. With sufficient diversity and quantity, we aim to show that these CF differences could span the spurious feature subspace.

Oracle CF pairs in (1) are infeasible to obtain in practice, but noisy CF pairs can be collected in certain scenarios. To address those, we propose an approximate spurious subspace matching method using the noisy CF pairs. Concretely, define the noisy counterfactual pair difference matrix as $\tilde{\Delta}_x := \left[\boldsymbol{x}_{e_1} - \boldsymbol{x}_{e'_1}, \ldots, \boldsymbol{x}_{e_k} - \boldsymbol{x}_{e'_k}\right] \in \mathbb{R}^{d \times k}$. Given noisy pairs, the matrix $\tilde{\Delta}_x$ has rank $k$. Enforcing the classifier to be orthogonal to the noisy CF differences, i.e., $\theta^\top \tilde{\Delta}_x = 0$, can lead to pathological outcomes. For example, if there are sufficient many CF pairs such that $k > |\mathcal{I}_{\mathcal{F}_{\mathcal{E}}}|$, then $\theta$ must be orthogonal to a larger subspace than the spurious feature subspace, leading to degraded performance. Thus, we propose to introduce NCM as follows:

$$\min_\theta \mathbb{E}_{(\mathbf{x},\mathbf{y}) \sim \mathbb{P}_{\text{train}}}[\ell(h(\mathbf{x};\theta), \mathbf{y})] \qquad \text{s.t.} \ \theta^\top \tilde{Q}_r = 0, \tag{2}$$

where $\tilde{Q}_r \in \mathbb{R}^{d \times r}$ denotes the space of left singular vectors corresponding to the $r$-truncated SVD of $\tilde{\Delta}_x$. With perfect counterfactuals, $\tilde{Q}_r$ correspond to the spurious subspace, and the classifier would be robust to changes in the spurious subspace. Because of the noise, a much delicate analysis is required to show that this approach improves robustness based on the diversity and quality of the noisy CF pairs. There are many efficient algorithms including reparameterization approach, projected gradient descent to solve the constrained problem (2), we refer the reader to Algorithm 1 in Appendix B for a detailed implementation.

## 4 THEORETIC GUARANTEES OF NCM FOR LINEAR MODELS

In this section, we provide theoretic guarantees of NCM (2) for both linear regression and logistic regression. Our study proceeds through four steps: (1) we decompose the test error into in-domain error and spurious subspace misalignment (Theorem 1); (2) we quantify this misalignment using Wedin's $\sin \Theta$ theorem (Wedin, 1972) (Corollary 3); (3) we characterize the out-of-domain risk under ERM (Corollary 2); and (4) we show that oracle CF pairs in (2) recover the optimal robust classifier (Corollary 4).

We consider both logistic regression and linear regression tasks, where the data generation processes are linear in both cases, differing only in the target variable y. In logistic regression, $g_\mathbf{y}(\mathbf{z})$ is a sign function composed with a linear function, while in linear regression, $g_\mathbf{y}(\mathbf{z})$ is a linear function. Given the data generating process, it is natural to consider the linear regressor $h_{\mathcal{E}}$ parameterized by $\theta$ to predict y from $\mathbf{x}$, and the optimally robust classifier $h_{\mathcal{E}}^*$ parameterized by $\theta^*$.

To quantify the deviation of a test domain $e^+ \in \mathcal{E}_{\text{test}}$ from the training domain $\mathcal{E}_{\text{train}}$, we introduce a conceptual random variable defined as $\mathbf{x}_{e^+ \to e} := g_\mathbf{x}(f_e(\mathbf{u}))$, where $\mathbf{u}$ is the same random variable shared by $\mathbf{x}_{e^+}$, as $\mathbf{x}_{e^+} = g_\mathbf{x}(f_{e^+}(\mathbf{u}))$. Therefore, for each realization of $\mathbf{u}$, the corresponding realization of $(\mathbf{x}_{e^+}, \mathbf{x}_{e^+ \to e})$, denoted as $(\boldsymbol{x}_{e^+}, \boldsymbol{x}_{e^+ \to e})$, forms an oracle CF pair. Note that the conceptual random variable $\mathbf{x}_{e^+ \to e}$ is used only for analysis and does not need to be observed in the training data. Given that the exogenous noise follows the distribution $\mathbb{P}_{\mathcal{U}}$ (cf. Definition 6), $\mathbf{x}_{e^+ \to e}$ follows the training domain distribution $p(\mathbf{x}_e)$. Building upon it, we introduce population second moment matrix and its SVD as follows $M_{e^+, \mathcal{E}_{\text{train}}} := \sum_{e \in \mathcal{E}_{\text{train}}} \mathbb{P}(e) \mathbb{E}_{\boldsymbol{u}}[(\mathbf{x}_{e^+} - \mathbf{x}_{e^+ \to e})(\mathbf{x}_{e^+} - \mathbf{x}_{e^+ \to e})^\top] = Q_{|\mathcal{I}(\mathcal{F}_{\mathcal{E}})|} \Lambda_{|\mathcal{I}(\mathcal{F}_{\mathcal{E}})|} Q_{|\mathcal{I}(\mathcal{F}_{\mathcal{E}})|}^\top$, where $Q_{|\mathcal{I}(\mathcal{F}_{\mathcal{E}})|}$ is the relevant spurious subspace for the latent SCM class $\mathcal{M}_{\mathcal{E}}$ where $\mathcal{E}$ contains the training domains and the $e^+$ test domain, and $\mathbb{P}(e)$ is the marginal distribution of environments in the training distribution. Note that the singular values greater than $|\mathcal{I}(\mathcal{F}_{\mathcal{E}})|$ are zero due to our spurious correlation assumption (Assumption 1). We have the following guarantee on NCM (2).

**Theorem 1** (Test-Domain Error Bound for NCM with Linear Models). *Assuming that the environments are defined by a class of linear spurious correlation latent SCMs $\mathcal{M}_{\mathcal{E}}$ (Assumption 1), the test-domain risk of any $\theta$ that satisfies the NCM constraint (2) for any test domain $\mathcal{M}_{e+} \in \mathcal{M}_{\mathcal{E}}$ is bounded as follows.*

a) *For logistic regression with log loss $\ell_{\mathrm{LL}}$, the following bound holds:*

$$\mathbb{E}_{(\mathbf{x}_{e+}, \mathbf{y}_{e+}) \sim \mathbb{P}_{\mathrm{test}}} \left[ \ell_{\mathrm{LL}}(\theta^\top \mathbf{x}_{e+}, \mathbf{y}_{e+}) \right] \leq \underbrace{\mathbb{E}_{(\mathbf{x}, \mathbf{y}) \sim \mathbb{P}_{\mathrm{train}}}[\ell_{\mathrm{LL}}(\theta^\top \mathbf{x}, \mathbf{y})]}_{\text{Term I: In-domain error}} + \underbrace{\|\theta\| \left\| \tilde{Q}_{r,\perp}^\top Q_{|\mathcal{I}(\mathcal{F}_{\mathcal{E}})|} \right\|_{\Lambda_{|\mathcal{I}(\mathcal{F}_{\mathcal{E}})|}}}_{\text{Term II: Spurious subspace misalignment}}.$$

b) *Similarly, for linear regression with squared error loss $\ell_{\mathrm{SE}}$, the following holds:*

$$\mathbb{E}_{(\mathbf{x}_{e+}, \mathbf{y}_{e+}) \sim \mathbb{P}_{\mathrm{test}}}[\ell_{\mathrm{SE}}(\theta^\top \mathbf{x}_{e+}, \mathbf{y}_{e+})] \leq 2 \underbrace{\mathbb{E}_{(\mathbf{x}, \mathbf{y}) \sim \mathbb{P}_{\mathrm{train}}}[\ell_{\mathrm{SE}}(\theta^\top \mathbf{x}, \mathbf{y})]}_{\text{Term I: In-domain error}} + 2\|\theta\|^2 \underbrace{\left\| \tilde{Q}_{r,\perp}^\top Q_{|\mathcal{I}(\mathcal{F}_{\mathcal{E}})|} \right\|_{\Lambda_{|\mathcal{I}(\mathcal{F}_{\mathcal{E}})|}}^2}_{\text{Term II: Spurious subspace misalignment}}.$$

*Furthermore, Term II in (a) and (b) can be bounded using the following:*

$$\left\| \tilde{Q}_{r,\perp}^\top Q_{|\mathcal{I}(\mathcal{F}_{\mathcal{E}})|} \right\|_{\Lambda_{|\mathcal{I}(\mathcal{F}_{\mathcal{E}})|}}^2 \leq \lambda_1(e^+)\mathrm{dist}^2(\tilde{Q}_s, Q_s) + \lambda_{s+1}(e^+), \tag{3}$$

*where $s := \min\{r, |\mathcal{I}(\mathcal{F}_{\mathcal{E}})|\}$ is the minimum of the user-specified $r$ and the dimension of the relevant spurious feature subspace, $\mathrm{dist}^2(Q, Q') := \|QQ^\top - Q'Q'^\top\|^2$ denotes the squared distance between subspaces (Chen et al., 2021), and $\lambda_1(e^+)$ and $\lambda_{s+1}(e^+)$ denote the largest and $(s+1)$-th largest eigenvalue of $M_{e+, \mathcal{E}_{\mathrm{train}}}$ respectively.*

See the appendix for proofs. The following comments are in order.

**(i) Error decomposition:** Observe that the test error due to spurious correlations can be categorized into two terms. Term I: *in-domain error* and Term II: *weighted spurious subspace misalignment*. In the extreme case, when we have full knowledge of the oracle counterfactual pairs and the the ambient true dimension $|\mathcal{I}(\mathcal{F}_{\mathcal{E}})|$, Term II vanishes, and thus, the test error reduced to in-domain error. Intuitively, Term II quantifies the weighted impact of subspace misalignment, where the weights are given by the eigenvalues of the true spurious subspace.

**(ii) Accuracy trade-off induced by $r$:** The second term in (3) captures the model misspecification error due to $r \neq |\mathcal{I}(\mathcal{F}_{\mathcal{E}})|$. In practice, $|\mathcal{I}(\mathcal{F}_{\mathcal{E}})|$ is unknown, so one may either overestimate or underestimate it through $r$. Our theory quantifies the impact explicitly within the bound:

(a) If we overestimate the spurious feature dimension, i.e., choose $r > |\mathcal{I}(\mathcal{F}_{\mathcal{E}})|$, then $s = |\mathcal{I}(\mathcal{F}_{\mathcal{E}})|$, and hence $\lambda_{s+1}(e^+) = \lambda_{|\mathcal{I}(\mathcal{F}_{\mathcal{E}})|+1}(e^+) = 0$. In this case, Term II vanishes, but Term I increases since a larger $r$ reduces the feasible region $Q_{r,\perp}$, resulting in greater in-domain error.

(b) If we underestimate the spurious feature dimension, i.e., choose $r < |\mathcal{I}(\mathcal{F}_{\mathcal{E}})|$, then $s = r$, and thus $\lambda_{s+1}(e^+) = \lambda_{r+1}(e^+) > 0$, which increases monotonically as $r$ decreases. Here, a smaller $r$ lowers Term I as the feasible region $Q_{r,\perp}$ is larger but simultaneously amplifies Term II due to the larger $\lambda_{r+1}(e^+)$.

This elegant trade-off is clearly observed in both synthetic and real-world datasets, as shown in the arc-shaped curves in Figure 2b and Figure 7, which illustrate how the model performance varies with different values of $r$.

One simple corollary of our theorem occurs in the extreme case when $r = 0$ such that NCM reduces to ERM. In this extreme case, we have that the subspace distance term is zero because $s = 0$ and $\lambda_{r+1}(e^+) = \lambda_1(e^+)$, which is typically large and reflects the spurious correlation captured by ERM. This yields the following test-domain error bound for ERM, which is novel to the best of the authors' knowledge. This clearly shows how ERM error increases with increasingly stronger shifts in the test domain.

**Corollary 2** (Test-Domain Error Bound for ERM with Linear Models). *Given the same assumptions as Theorem 1, the test-domain error for ERM for logistic regression is bounded by:*

$$\mathbb{E}_{(\mathbf{x}_{e+}, \mathbf{y}_{e+}) \sim \mathbb{P}_{\mathrm{test}}} \left[ \ell_{\mathrm{LL}}(\theta^\top \mathbf{x}_{e+}, \mathbf{y}_{e+}) \right] \leq \mathbb{E}_{(\mathbf{x}, \mathbf{y}) \sim \mathbb{P}_{\mathrm{train}}}[\ell_{\mathrm{LL}}(\theta^\top \mathbf{x}, \mathbf{y})] + \|\theta\| \sqrt{\lambda_1(e^+)},$$

*and similarly for linear regression.*

Next, we leverage Wedin's $\sin \Theta$ theorem Wedin (1972) to further characterize the spurious subspace misalignment term (Term II in Theorem 1) based on the counterfactual noise $\varepsilon := \tilde{\Delta}_{\mathbf{x}} - \Delta_{\mathbf{x}}$, where $\Delta_{\mathbf{x}}$ denotes the corresponding perfect counterfactuals. For this corollary, we assume that the oracle counterfactuals corresponding to the observed counterfactuals spans the entire spurious subspace. Thus, we can isolate the effect of noisy counterfactuals compared to oracle counterfactuals. This shows that if the counterfactuals are diverse enough (i.e., they span the spurious subspace) and have bounded noise levels, then we can improve test-domain performance.

**Corollary 3** (Test-Domain Bound in Terms of Counterfactual Noise). *Instate the setting in Theorem 1. Further, assume that the oracle counterfactual difference $\Delta_{\mathbf{x}}$ has a rank equal to the spurious feature subspace, i.e., $\mathrm{rank}(\Delta_{\mathbf{x}}) = |\mathcal{I}(\mathcal{F}_\mathcal{E})|$, and suppose the noise matrix $\varepsilon := \tilde{\Delta}_{\mathbf{x}} - \Delta_{\mathbf{x}}$ (with singular values of $\Delta_{\mathbf{x}}$ denoted by $\sigma_j$) satisfies $\sigma_1 \leq (1 - 1/\sqrt{2})(\sigma_s - \sigma_{s+1})$ where $s = \min\{r, |\mathcal{I}(\mathcal{F}_\mathcal{E})|\}$. Then, for any $\theta$ that satisfies the NCM constraint (2) and any test domain satisfying $\mathcal{M}_{e^+} \in \mathcal{M}_\mathcal{E}$, the following holds for logistic regression:*

$$\mathbb{E}_{(\mathbf{x}_{e^+}, \mathbf{y}_{e^+}) \sim \mathbb{P}_{\text{test}}} \left[ \ell_{\mathrm{LL}}(\theta^\top \mathbf{x}_{e^+}, \mathbf{y}_{e^+}) \right] \leq \mathbb{E}_{(\mathbf{x}, \mathbf{y}) \sim \mathbb{P}_{\text{train}}} [\ell_{\mathrm{LL}}(\theta^\top \mathbf{x}, \mathbf{y})] + \|\theta\| \left( \frac{2\sqrt{\lambda_1(e^+)}}{\sigma_s - \sigma_{s+1}} \|\varepsilon\| + \sqrt{\lambda_{s+1}(e^+)} \right),$$

*and similarly for linear regression.*

**Choice of Clean $\Delta_{\mathbf{x}}$.** It is critical to note that we do not make any assumptions about the noise except that $\sigma_1 \leq (1 - 1/\sqrt{2})(\sigma_s - \sigma_{s+1})$.[2] Thus, given any observed noisy counterfactuals, we could actually choose any such real counterfactual matrix that satisfies this condition, i.e., we could theoretically choose an oracle counterfactual matrix $\Delta_{\mathbf{x}}$ that spans the spurious space and minimizes the bound. Therefore, this result shows that as long as the counterfactuals are reasonably diverse (i.e., they span the spurious subspace) and they are not too noisy, our NCM approach will improve the robustness to spurious correlation.

**A Small Number of Perfect Counterfactual Pairs Is Sufficient.** Additionally, we note that a simple corollary of this result is that by choosing $s \geq |\mathcal{I}(\mathcal{F}_\mathcal{E})|$ with perfect counterfactuals (i.e., no noise), then the test-domain error will equal the train domain error. Furthermore, to satisfy the rank condition, theoretically we only need $|\mathcal{I}(\mathcal{F}_\mathcal{E})|$ noiseless pairs whose differences are linearly independent of each other. This emphasizes that in the ideal case only a small number of diverse pairs are needed. In contrast to the IRM requires $|\mathcal{E}_{\text{train}}| \geq \mathcal{I}(\mathcal{F}_\mathcal{E})$ (Rosenfeld et al., 2020). If there is noise, then more pairs will be needed but we expect that still only a small number is needed to improve robustness. We further observe this empirically (cf. Section 5).

## 5 EMPIRICAL EVALUATION

In this section, we present experiments on both synthetic and real world datasets. **(i)** The synthetic dataset is used to validate the theoretical results. We evaluate robustness using both noisy and oracle CF pairs, and confirm the few shot counterfactual pairs requirement of the CF pairs and the spurious feature dimension (cf. Corollary 3). We also validate the trade-off effect of $r$ (cf. Theorem 1 (ii)). **(ii)** Beyond synthetic data, we evaluate NCM (2) via linear probing on a frozen CLIP model (Radford et al., 2021) across three real world datasets: ColoredMNIST, PACS, and Waterbirds. While CLIP already demonstrates strong zero shot transfer (Radford et al., 2021), NCM (2) further improves robustness to spurious correlations over CLIP. **(iii)** We compare three matching strategies: random matching, nearest neighbor matching (Mahajan et al., 2021), and NCM (2), and show the superiority of CF based approaches. Details on data generation, validation, and hyperparameter tuning are provided in Appendix D.2. All experiments report in-domain validation in the main text, with additional results including oracle-validation, hyperparameter sensitivity, and experiments with traditional deep models as well as a result from the additional PACS dataset are provided in Appendix E.

---

[2]Such a technical condition results from Wedin's $\sin \Theta$ theorem (Wedin, 1972). In reality, we found that the performance of NCM still outperforms ERM even when such a condition is not satisfied (cf. Section 5) and could be benefited by utilizing more pairs (cf. Figure 2a).

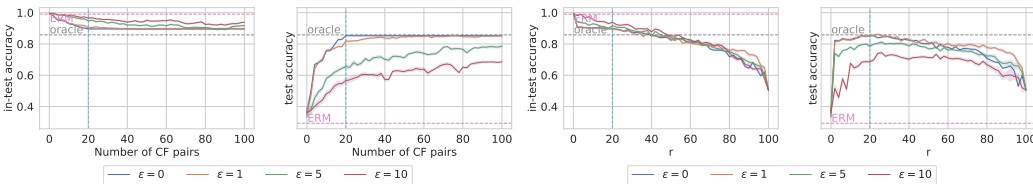

(a) Acc vs. $k$ under different noise, $r = \min(k, 20)$.  (b) Acc vs. $r$ under different noise, $k = 100$.

Figure 2: Result on the synthetic dataset with NCM. We report both in-domain test accuracy (in-test accuracy) and test domain accuracy (test accuracy). We choose $m = 100$ and $|\mathcal{I}(\mathcal{F}_{\mathcal{E}})| = 20$ (denoted by vertical dash line). The horizontal lines represent the ERM accuracy and oracle accuracy (ERM train on test domain). The vertical line at 20 denotes $\mathcal{I}(\mathcal{F}_{\mathcal{E}})$. $\varepsilon = 0$ means oracle CF pairs. The solid curves represent the mean over 10 runs with shaded regions indicating standard deviations.

**Synthetic Experiments.** The results indicate that (a) *oracle accuracy:* NCM (2) achieves oracle-level accuracy under small noise of CF pairs ($\varepsilon = 0, 1$), as if the model were trained directly on the test domain (see Figure 2a). (b) *Few-shot CF pairs:* we observe that only $k = \mathcal{I}(\mathcal{F}_{\mathcal{E}}) = 20$ oracle CF pairs are required to correctly identify the spurious space, achieving the best possible performance. When the noise is small ($\varepsilon = 1$), the performance remains optimal. However, as the noise become larger ($\varepsilon = 5, 10$), the performance degrades, as predicted by Theorem 1. (c) *Trade-off effect of $r$:* we fix the number of noisy CF pairs as $k = 100$ and evaluate the effect of varying the subspace rank $r$ on test accuracy (see Figure 2b) under different noises. For small noise levels ($\varepsilon = 0, 1$), we observe monotonically decreasing in-domain test accuracy and the arch shape test accuracy curves (cf. Theorem 1 (ii)), showing the trade-off effect of $r$. The best performance achieves when $r \approx \mathcal{I}(\mathcal{F}_{\mathcal{E}})$. In this case of large noise ($\varepsilon = 5, 10$), the noise makes truncated SVD decomposition more prone to preserving some spurious features while removing some invariant features, so a slightly aggressive selection of $r$ yields better results.

Table 1: Main Results with in-domain validation. Results with oracle validation can be found in Table 3, Table 4, and Table 5 in the appendix. "WG" represents worst group.

|  | Data | Model | ColoredMNIST | | Waterbirds | |
|---|---|---|---|---|---|---|
|  |  |  | in acc | test acc | in acc | wg acc |
| ERM (CLIP | DG | Probing | 0.852 | 0.093 | 0.885 | 0.781 |
| IRM | DG | Probing | 0.799 | 0.118 | 0.838 | 0.707 |
| REx | DG | Probing | 0.797 | 0.121 | 0.891 | 0.617 |
| GroupDRO | DG | Probing | 0.798 | 0.127 | 0.906 | 0.684 |
| Fish | DG | Probing | 0.798 | 0.118 | 0.900 | 0.744 |
| SWAD | DG | Probing | 0.800 | 0.113 | - | - |
| LISA | DG | Probing | 0.705 | **0.000** | 0.904 | 0.722 |
| MatchDG | 1NN | CNN | 0.698 | 0.361 | 0.970 | 0.080 |
| MatchDG | 1NN | Finetune | 0.850 | 0.181 | 0.920 | 0.112 |
| MatchDG | random | Probing | 0.799 | 0.120 | 0.793 | 0.009 |
| MatchDG | 1NN | Probing | 0.789 | 0.217 | 0.886 | 0.411 |
| MatchDG | clean | Probing | 0.793 | 0.181 | 0.906 | 0.536 |
| NCM | random | Probing | 0.794 | 0.176 | 0.804 | 0.269 |
| NCM | 1NN | Probing | 0.736 | 0.649 | 0.892 | 0.521 |
| NCM | clean | Probing | 0.740 | **0.693** | **0.864** | **0.812** |
| random guess |  |  | 0.500 | 0.500 | - | - |
| ERM oracle |  |  | 0.735 | 0.730 | - | - |
| theory oracle |  |  | 0.750 | 0.750 | - | - |

**Real-world Dataset.** We present the results of NCM (2) on two representative real-world datasets. ColoredMNIST (Arjovsky et al., 2019) is a semi-synthetic dataset, but is widely recognized as difficult due to strong *accuracy on the inverse line effect* (Gulrajani and Lopez-Paz, 2021; Salaudeen et al., 2024). Waterbirds-CF is a highly imbalanced dataset, where the minority group consists of only 240 samples. We construct CF pairs by matching these with 240 randomly selected samples from the majority group of original (See Appendix D.1 for detail). This dataset is used to highlight the robustness of our method in this domain-imbalance scenario.

We summarize our results in Table 1 and give more detailed tables in the appendix. On ColoredM-NIST result, our result shows that NCM probing on CLIP pretrained model (2) performs well on both in-domain and oracle-validation (cf. Table 3), achieving test domain accuracies of 69.3% and 71.4%, respectively, nearly matching the ERM oracle accuracy of 73%, demonstrating the effectiveness of NCM (2). The performance difference between two validation methods are only 2%, indicating that NCM (2) is less sensitive to hyperparameter tuning. This stands in sharp contrast to other algorithms such as ERM, IRM, GroupDRO, Fish, and REx, which only achieve around 10% accuracy with in-domain validation and 20%-66% with oracle validation except for LISA which

achieves 69.3% on both validation methods. Our results show that NCM (2) with noisy CF pairs achieves 81.2% worst-group accuracy, outperforming the best baseline (ERM) by 3.1% using in-domain validation. In contrast, all other methods underperform ERM. We further include the oracle validation and other baseline methods with CLIP on oracle validation. It also achieves 86% accuracy using oracle validation (see Table 4), outperforming the best probing method on CLIP by 3.3%. We further include finetuning MatchDG with iterative matching as well as the ResNet50 end-to-end training for comparison with iterative matching. Due to the non-linear backbone and the existence of noises, the MSE constraint cannot effectively find the correct invariant subspace thus suffers from the suboptimal results. Our observation are as follows: First, NCM (2) consistently outperforms all baselines across these datasets. Second Random pairing and 1 Nearest Neighbor (1NN) pairing perform well on ColoredMNIST, but fail on Waterbirds. On ColoredMNIST, invariant features are inherently similar across samples, allowing even random pairing to produce reasonable noisy CF pairs. In contrast, Waterbirds exhibit greater variability in the features making it difficult for 1NN to find meaningful counterfactual matches.

## 6 RELATED WORKS

*Data augmentation and generation:* Data augmentations can be seen as simple-to-generate counterfactual pairs, where the augmentations implicitly encode knowledge about desired invariances. For example, standard functions like rotation, scaling, and noise addition suggest that such transformations should not alter the predicted class (Honarvar Nazari and Kovashka, 2020; Shorten and Khoshgoftaar, 2019). More sophisticated strategies follow this principle; LISA, for instance, is a Mixup-inspired method that learns domain-invariant predictors through intra-label and intra-domain mixing to encourage the model to respect class boundaries (Yao et al., 2024). DIRT (Nguyen et al., 2021) suggests using StarGAN (Choi et al., 2018) to generate paired samples, while other work has used ComboGAN (Anoosheh et al., 2018) to generate new data (Rahman et al., 2019). From our perspective, these generated samples can be seen as complex, class-preserving data augmentations to estimate pairs regarding some type of invariances. Our work provides a causal language and theoretical guarantee for this approach.

*Distribution or sample matching in addressing spurious correlations:* Invariant Risk Minimization (IRM) (Arjovsky et al., 2019) aim to mitigate spurious correlations by learning domain-invariant representations. Despite their theoretical appeal, IRM-based approaches often under perform in practice, prompting several works to analyze and refine them (Rosenfeld et al., 2020; Krueger et al., 2021; Ahuja et al., 2022). Beyond distribution matching, MatchDG (Mahajan et al., 2021) introduces an iterative sample-level matching objective that aligns representations across domains in latent space. Our method similarly employs sample-wise matching but crucially, we provide a theoretical robustness guarantee and deeper exploration on the properties of these pairs.

*Causal inference seeking invariant predictor for robustness:* The goal of domain generalization in a causal perspective is to find a representation $\Phi$ of $\mathbf{x}$ such that $y \perp e|\Phi(\mathbf{x})$. Different approach to induce $\Phi$ has been heavily explored. Most of the causal inference type of work focusing on observable causal variables Magliacane et al. (2018) proposes to find subset of causal variable $\Phi(\mathbf{x})$ in $\mathbf{x}$, where $\mathbf{x}$ is the set of observable causal variables and $\Phi(\mathbf{x}) \subset \mathbf{x}$, such that $y \perp e|\Phi(\mathbf{x})$ holds. Subbaswamy et al. (2019) considers the graph surgery estimator that finding the stable estimator by removing unstable mechanism from the joint factorization. However, it is extremely hard when the causal variables are latent.

## 7 CONCLUSION AND DISCUSSION

We address spurious correlations from a data-centric view, showing that introducing (noisy) counterfactual pairs during training improves model robustness. This mirrors classical supervised learning, where labels guide models toward target concepts without formal definitions; similarly, invariant pairs implicitly identify and mitigate spurious features. One challenge of our method is obtaining counterfactual pairs. While straightforward in tasks like object classification (e.g., using image editing for spurious features, as shown in the Waterbirds dataset), it is more complex in fields like medical imaging, requiring expert involvement. However, experts can now help by creating or validating a few high-quality counterfactuals to improve robustness suggested by our findings.

## REPRODUCIBILITY STATEMENT AND ETHICS STATEMENT

**Reproducibility Statement**  A link to an anonymous downloadable source code can be found in the appendix; We further include the details to reproduce the results in Appendix D.

**Ethic State**  The authors have read and adhere the ICLR Code of Ethics.

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

# APPENDIX

The code is available at https://anonymous.4open.science/r/NCM-A35E.

## THE USE OF LARGE LANGUAGE MODELS (LLMS)

In the preparation of this manuscript, authors utilized the Large Language Model (LLM) to assist in two capacities: for brainstorming initial conceptual approaches to theoretical proofs and for refining the text for grammatical accuracy and clarity. The authors are fully responsible for all substantive content and the final scientific conclusions.

## A EXPANDED EXPLANATION

In this section, we include additional discussions about the causal model, assumptions, data availability, etc.

### A.1 EXPANDED EXPLANATION OF LATENT CAUSAL MODEL

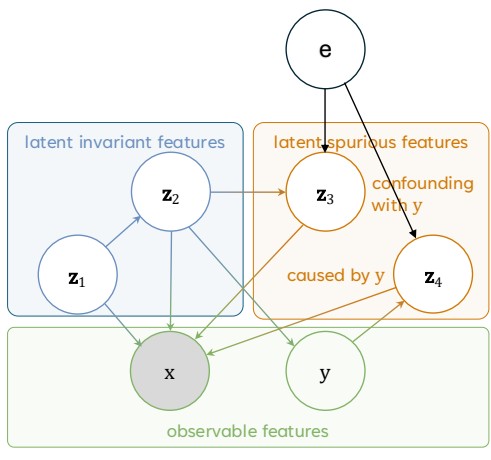 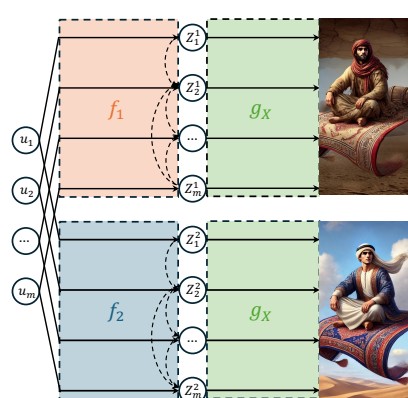

Figure 3: Illustration of the latent causal model. The ancestors of the target y are $\mathbf{z}_1, \mathbf{z}_2$, which are assumed to be invariant across domains (see Assumption 1). On the other hand, $\mathbf{z}_3, \mathbf{z}_4$ are spurious features. To be specific, $\mathbf{z}_3$ is confounded with y, and $\mathbf{z}_4$ is descendant of y. Because they are not ancestors of y, thus they are spurious.

Figure 4: An illustration of oracle counterfactual pairs represented by our model, where $f_1$ and $f_2$ are two SCMs' solution function for domain 1 and domain 2, $g_{\mathbf{x}}$ is the observation function from $\mathbf{z}$ to $\mathbf{x}$. In this figure, we do not plot the prediction target y and correspondingly $g_y$.

Figure 3 is an illustration of the proposed latent causal model, which satisfies the conditions in Definition 6 and Assumption 1. We first note that this is a fairly common assumption in the field (e.g., Rosenfeld et al. (2020); Arjovsky et al. (2019)). We also suggest that this is not a significant limitation. Suppose we had a graph where directly causes, i.e., $y = g_y(u_y, x, z_{\text{Pa}(y)})$. We could then create a new graph with a new latent node equal to $x$, i.e., $z_x = x$ and change $g_x(z_1, z_2, \ldots, z_x, \ldots,) = z_x = x$ but now $g_y$ would only depend on latent variables. Thus, because we already deal with complex latent variables, this is a not a limitation but more syntactic.

The domain counterfactuals encode crucial information about the underlying data generation mechanisms and help avoid reliance on features that are spuriously correlated with labels in the training dataset. Oracle counterfactual pairs are samples living in two different causal worlds that shares the same exogenous noise. In the example of Figure 4, two Aladdin images are oracle counterfactual pairs where the intervention variable is "wealth" represented by some latent variable $z_i$. $g_{\mathbf{x}}$ encodes the causal information to images.

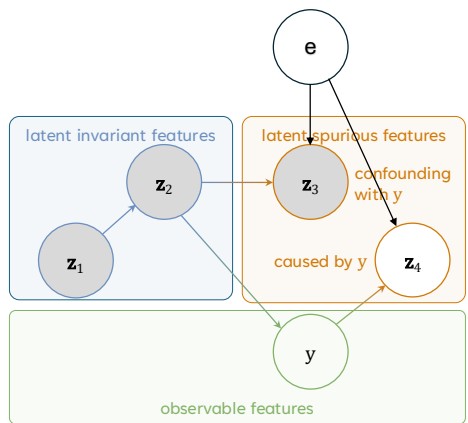 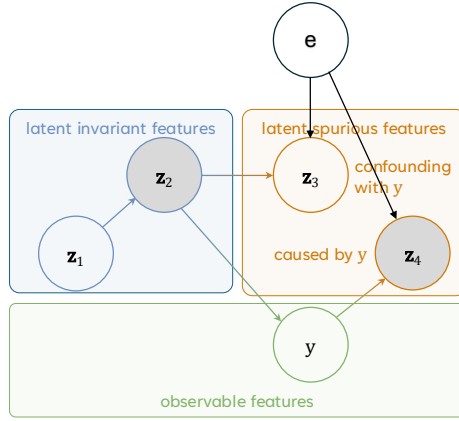

(a) With known causal fes, we have a robust predictor $\mathbf{y} \perp \mathbf{e}|\mathbf{z}_1, \mathbf{z}_2, \mathbf{z}_3$. Yet, because of the graph, we know $p(\mathbf{y}|\mathbf{z}_1, \mathbf{z}_2, \mathbf{z}_3) = p(\mathbf{y}|\mathbf{z}_1, \mathbf{z}_2)$. This suggests that including $\mathbf{z}_3$ is safe but the optimal predictor will be constant to $\mathbf{z}_3$ due to the conditional independence, thus does not violate our lemma.

(b) With known causal variables, we have $\mathbf{y} \not\perp \mathbf{e}|\mathbf{z}_2, \mathbf{z}_4$. Thus, using $\mathbf{z}_4$ is not robust, which is the same as previous results.

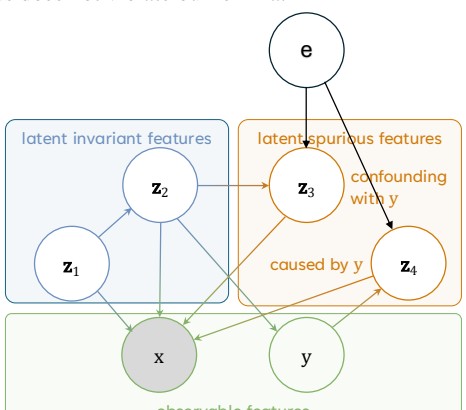 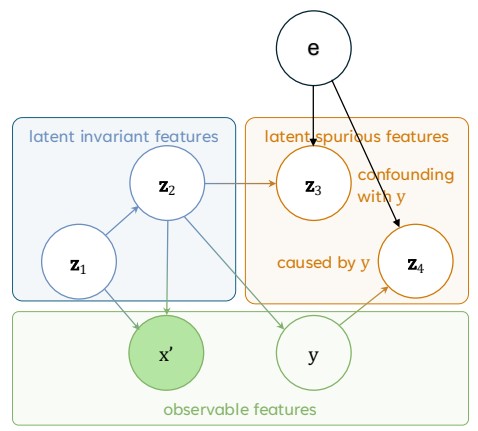

(c) For spurious correlation, we have $\mathbf{y} \perp \mathbf{e}$, but because we condition on the collider $\mathbf{x}$, we have $\mathbf{y} \not\perp \mathbf{e}|\mathbf{x}$, leading to non-robust predictor. Even when we further hypothetically condition on any subset of the latent variables $\mathbf{z}' \subset \{\mathbf{z}_1, \mathbf{z}_2, \mathbf{z}_3, \mathbf{z}_4\}$, we still have $\mathbf{y} \not\perp \mathbf{e}|\mathbf{z}'$, i.e., a non-robust predictor..

(d) Our proposed method can be seen as post-processing intervention $\phi$ on $\mathbf{x} = g(\mathbf{u_x}, \mathbf{z}_1, \mathbf{z}_2, \mathbf{z}_3, \mathbf{z}_4)$ such that $\mathbf{x}' := \phi(\mathbf{x}) = \phi(g(\mathbf{u_x}, \mathbf{z}_1, \mathbf{z}_2, \mathbf{z}_3, \mathbf{z}_4)) = g'(\mathbf{u_x}, \mathbf{z}_1, \mathbf{z}_2)$ for some $g'$ that only depends on $\mathbf{u_x}$ and $\mathbf{z}_1$. The post-processing function $\phi$ can be seen as forming a new intervened SCM with certain incoming edges removed. Invariant pairs provide additional signal to find such $\phi$. In the linear case, we simply project out the spurious feature by truncated SVD.

Figure 5: Illustration of robustness prediction corresponding to non-latent and latent causal variables.

From a causal perspective, the key of robustness to spurious correlation is getting independence of $\mathbf{y}$ and $\mathbf{e}$ given some conditioning statements. If the causal $\mathbf{z}$ variables are observable, then one could use a d-separation criteria to realize that $\mathbf{z}_1$, $\mathbf{z}_2$ and $\mathbf{z}_3$ is a d-separating set for $\mathbf{e}$ and $\mathbf{y}$ as in Figure 5a. In this simplified scenario, conditioning on $\mathbf{z}_4$ would not be robust as illustrated in Figure 5b. However, when conditioning on $\mathbf{x}$, $\mathbf{y}$ and $\mathbf{e}$ are dependent because of the collider effect of $\mathbf{x}$ and that $\mathbf{z}_4$ is a descendant of $\mathbf{y}$ as illustrated in Figure 5c. Our method is able to overcome this limitation by being viewed as a postprocessing intervention on $\mathbf{x}$ that removes the edges from the spurious features to $\mathbf{x}$. This enables $\mathbf{e}$ and $\mathbf{y}$ to be independent. From this perspective, our approach learns this postprocessing function $\phi$ based on counterfactual pairs.

## A.2 AVAILABILITY OF INVARIANT PAIRS

While invariant pairs can be hard to acquire, we argue that it is feasible and practical to obtain in certain scenarios.

For certain applications, obtaining such CF pairs are both possible and effective. Table 2 from the introduction summarizes a range of cases where there could be enough implicit knowledge of spurious correlations to collect them. We further outline these levels in detail below.

Table 2: An illustrative taxonomy of scenarios from explicit knowledge to no knowledge of spurious correlations.

|         | knowledge on spurious features | pair data acquisition |
| ------- | ------------------------------ | --------------------- |
| level 3 | explicit knowledge             | model constraint      |
| level 2 | soft expert knowledge          | sample editing        |
| level 1 | implicit assumed               | CF pair collection    |
| level 0 | no knowledge                   | -                     |

*Level 3 - Explicit knowledge:* In some scientific settings, spurious correlations can be coded as an explicit and mathematical modeling constraint. For example, SchNet (Schütt et al., 2018) builds molecule symmetries and invariance directly into the model structure. This case is straightforward but does not hold in general, so we do not consider it in our work.

*Level 2 - Domain expert "soft" knowledge of spurious features:* In some applications, domain experts can articulate which features are irrelevant, even if they cannot encode this knowledge as model constraints. For example, an x-ray technician knows that certain medical equipments should not affect their diagnosis of cancer or not (Zech et al., 2018; Oakden-Rayner et al., 2020). In this case, CF pairs can be either manually curated (via image editing or generative models) or collected (e.g., by obtaining paired x-rays with and without fluid lines). Simple image augmentation techniques like rotations, flips or color distortions may also fall under this category as they implicitly encode spurious features that are assumed to not affect the downstream tasks (like ColoredMNIST experiment (cf. Section 5)).

*Level 1 - Implicit knowledge:* At this level, the only differences between domains are assumed to be spurious features because of application-specific knowledge, but domain experts may not know the spurious features a priori. As one example, the differences between data coming from two similar microscopes can be assumed to be spurious since the measurement effects should not affect the underlying physical phenomena of interest. In this case, it is feasible to collect a small number of counterfactual pairs by measuring a small number of samples with both microscopes.

*Level 0 - No knowledge:* Without any hints or assumptions about spurious features as in levels 1-3, making a model robust to spurious features is likely infeasible. To illustrate, consider a simple causal structure without any knowledge on (latent) spurious features: $\mathbf{z}_1 \to \mathrm{y} \to \mathbf{z}_2$ where only $\mathbf{z}_1$ is invariant. Without any knowledge, there is no information to distinguish between invariant feature $\mathbf{z}_1$ and spurious feature $\mathbf{z}_2$. Moreover, if $\mathbf{z}_2$ is more strongly correlated to y or related to y that is easier to extract from inputs $\mathbf{x}$, models are prone to shortcut learning (Hermann et al., 2024), the model prediction will rely heavily or nearly solely on $\mathbf{z}_2$.

We specifically target the hard and feasible levels 1 and 2, and suggest that in certain cases these pairs could feasibly be collected or created either via manual editing or generative AI tools (Rombach et al., 2022; Betker et al., 2023). It seems that our method requires additional domain knowledge compared to the standard DG setting. We claim this is an *alternative* form of domain knowledge, as the standard DG setting also requires domain knowledge, as it is encoded in the multi-domain data collection (see Appendix A.4 for further explanation). Noticing that this CF pairs acquisition are still costly, so we ask: *if we have $k$ estimated CF pairs with noise $\varepsilon$, what kind of robustness guarantee can we get?*

## A.3 DISCUSSION ON SAMPLE COMPLEXITY

We wanted to further clarify the difference between the data requirements of IRM and NCM by considering the data scaling requirements for a certain number of intervened/spurious features, i.e., $|\mathcal{I}(\mathcal{F}_\mathcal{E})|$. For this comparison, we will assume an theoretically ideal setting with an infinite number of samples for each domain, but a finite number of perfect counterfactual pairs, where the true data-generating process is compatible with logistic loss.

IRM (Arjovsky et al., 2019): As proven in a prior study (Rosenfeld et al., 2020, Corollary 5.2), achieving optimal invariant predictors with IRM requires the number of training domains e to be

greater than the number of spurious feature dimensions, i.e., $e > |\mathcal{I}(\mathcal{F}_\mathcal{E})|$. And this requirement is true even if there is an infinite number of samples in each of the domains.

Noisy Counterfactual Matching (NCM): Our method, NCM, requires the number of linear independent invariant pairs, $k$, to be greater than the spurious feature dimension to achieve optimal invariant predictors, as shown in our paper's Corollary 4, i.e., $k \geq |\mathcal{I}(\mathcal{F}_\mathcal{E})|$. Linear independence could be satisfied if we assume full rank exogenous noise and soft intervention across domains, which are both common assumptions that are easy to satisfy.

It is important to note that the data-generating process for IRM is a special case of the structural causal model (SCM) that our NCM uses. (Specifically, while IRM does not account for the ancestors of the target variable e, it does permit some descendants of y to be unintervened, which are safe features for prediction based on d-separation.)

This distinction of the data requirement has significant practical implications, particularly in high-dimensional applications where the spurious feature dimension could be large: IRM would require the number of domains e to scale with the spurious feature dimension, which may be infeasible or too costly in practice (e.g., x-ray machine example). Our NCM, on the other hand, only requires the number of counterfactual pairs $k$ to scale with the spurious feature dimension, which may be significantly more feasible (e.g., x-ray machine example).

### A.4 DISCUSSION OF IMPLICIT DOMAIN KNOWLEDGE REQUIREMENT

Nearly all methods require domain knowledge to some extends. For instance, IRM implicitly uses expert's knowledge based on the specification of the domain labels. In IRM, domain labels are by definition the way of specifying what the predictions should be invariant to. Another example of using expert knowledge is data augmentations (See section 6 in our paper). While it seems that data augmentations don't use any domain knowledge, they actually implicitly use the domain knowledge that "the predictions should not change under this augmentation" (e.g., small rotations or color distortions). While not explicit, these data setups actually incorporate domain knowledge. As a concrete example, take the Rotated MNIST dataset. If the goal is to predict the digit, then rotation can be a domain label. However, if the goal is to predict the rotation, then the digit can be a domain label. Thus, knowledge about the task and what is irrelevant is key for even defining what parts of the data can be considered domain labels. We argue that the expert knowledge required for validating or creating domain counterfactuals is similar in spirit to the expert knowledge for defining domain labels or data augmentations. They are implicit ways of incorporating expert knowledge.

While our data setup differs from standard domain generalization tasks, we argue that expert knowledge is not required to employ the learning algorithm itself, but rather to construct the appropriate dataset. One approach is to use standard domain generalization methods like IRM, which require labeled data from multiple domains. In contrast, we present an alternative approach that requires only possibly noisy invariant pairs and labeled data from a single domain. Note that in our setting, the invariant pairs do not need to be labeled (for example, different x rays of the same patient even if the diagnosis is unknown). In all cases, whether using domain labels as in IRM, data augmentation as in LISA, or counterfactual pairing as in NCM, our algorithms can be generically applied given the appropriate data constructed through expert knowledge.

## B PRACTICAL ALGORITHM

Algorithmically, to ensure the learned model $\theta$ orthogonal to the spurious feature subspace estimated by r-truncated SVD decomposition of the noisy CF pairs, we consider reparameterization approach that projects the samples $\mathbf{x}$ onto the orthogonal complement of $\tilde{Q}_r$ (i.e., $I - \tilde{Q}_r \tilde{Q}_r^\top$) and then trains an unconstrained classifier (See Algorithm 1). This ensures the classifier only use the invariant component of $\mathbf{x}$ to predict $y$. This approach processes the data using the CF pairs before optimization, thus allows simple optimization approach. Other algorithm like projected gradient descent method could also be used here, and we expect it would have similar results, but we do not explore it further.

---

**Algorithm 1** Noisy Counterfactual-Matching

---

**Input:** Training Dataset $\mathcal{D}_{\text{train}}$; pair difference matrix $\tilde{\Delta}_{\mathbf{x}} \in \mathbb{R}^{d \times k}$; truncated SVD size $r$; epochs $T$; step size $\eta$; batch size $B$.

    *// Phase I: Find projection matrix to remove estimated spurious subspace $\tilde{Q}_r$.*
    $\tilde{Q}_r, \tilde{\Sigma}_r, \tilde{V}_r^\top = \text{TruncatedSVD}(\tilde{\Delta}_{\mathbf{x}}, r)$
    $P = I - \tilde{Q}_r \tilde{Q}_r^\top$
    *// Phase II: Gradient descent with preprocessing.*
    **for** $t = 1, 2, \ldots, T$ **do**
        **for** sample mini-batch $\{(\boldsymbol{x}_i, \boldsymbol{y}_i)\}_{i=1}^B \subset \mathcal{D}_{\text{train}}$ **do**
            $\theta \leftarrow \theta - \eta \nabla \frac{1}{B} \sum_{i=1}^B \ell(h(P\boldsymbol{x}_i; \theta), \boldsymbol{y}_i)$,
        **end for**
    **end for**
**Output** $\theta$

---

## C PROOFS

### C.1 PROOF OF PROPOSITION 1

The proof is based on the idea that the optimally robust classifier cannot vary w.r.t. the non-ancestors of y. Intuitively, if it does, then there exists an (adversarial) environment that can make the objective high. Given this invariance of the optimally robust predictor, it is simple to see that counterfactuals are indeed invariant pairs. For theoretic clarity, we will present the lemma that the optimally robust predictor is invariant to non-ancestors of y. Then, we will prove the proposition given this lemma and finally give the proof of the lemma.

For simplicity of notation in this section, we will let $\mathbf{z}_1 := \mathbf{z}_{\text{Anc}(y)}$ denote the latent $\mathbf{z}$ variables that are ancestors of y and let $\mathbf{z}_{2,e} := \mathbf{z}_{\mathcal{Z} \setminus \text{Anc}(y),e}$ denote variables that are not ancestors. Note that $\mathbf{z}_{2,e}$ depends on the environment $e$ but $\mathbf{z}_1$ does not depend on $e$ by our spurious correlation latent SCM class assumption (cf. Assumption 1).

**Lemma 1** (Optimally Robust Predictor is Invariant to Non-Ancestors). *Any optimally robust predictor for a spurious correlation latent SCM class must be constant w.r.t. the non-ancestors of* y *almost everywhere, i.e., for almost all $\boldsymbol{u}_{\mathbf{x}}$ and $\boldsymbol{z}_1$, there exists a constant $c_{\boldsymbol{u}_{\mathbf{x}}, \boldsymbol{z}_1}$ w.r.t. $\mathbf{z}_{2,e}$ such that $h_{\mathcal{E}}^*(g_{\mathbf{x}}(\boldsymbol{u}_{\mathbf{x}}, \boldsymbol{z}_1, \boldsymbol{z}_{2,e})) = c_{\boldsymbol{u}_{\mathbf{x}}, \boldsymbol{z}_1}$ almost everywhere.*

**Proposition 1** (Spurious Counterfactuals are Invariant Pairs). *Given a spurious correlation latent SCM class $\mathcal{M}_{\mathcal{E}}$ and a strictly convex loss function $\ell$, any observed counterfactual pair $(\boldsymbol{x}_e, \boldsymbol{x}_{e'})$ between $\mathcal{M}_e \in \mathcal{M}_{\mathcal{E}}$ and $\mathcal{M}_{e'} \in \mathcal{M}_{\mathcal{E}}$ will be an invariant pair w.r.t. the optimally robust classifier $h_{\mathcal{E}}^*$ based on $\ell$ induced by the domain distributions $\{\mathbb{P}_e\}_{e \in \mathcal{E}}$ almost surely, i.e., $h_{\mathcal{E}}^*(\boldsymbol{x}_e) = h_{\mathcal{E}}^*(\boldsymbol{x}_{e'})$.*

*Proof of Proposition 1.* By the definition of spurious counterfactual pairs, we know that $\boldsymbol{x}_e$ and $\boldsymbol{x}_{e'}$ must come from the same exogenous noise $\mathbf{u}$. And because we assume that no causal mechanism for ancestors is intervened, this means that there exists $(\mathbf{u}_{\mathbf{x}}, \mathbf{z}_1, \mathbf{z}_{2,e}, \mathbf{z}_{2,e'})$ such that $g_{\mathbf{x}}(\mathbf{u}_{\mathbf{x}}, \mathbf{z}_1, \mathbf{z}_{2,e}) = \boldsymbol{x}$ and $g_{\mathbf{x}}(\mathbf{u}_{\mathbf{x}}, \mathbf{z}_1, \mathbf{z}_{2,e'}) = \boldsymbol{x}'$. Because of Lemma 1, we know that the optimally robust predictor $h_{\mathcal{E}}^*$ must be invariant to changes in $\mathbf{z}_{2,e}$ values. Therefore, we have:

$$h_{\mathcal{E}}^*(\boldsymbol{x}) = h_{\mathcal{E}}^*(g_{\mathbf{x}}(\mathbf{u}_{\mathbf{x}}, \mathbf{z}_1, \mathbf{z}_{2,e})) = h_{\mathcal{E}}^*(g_{\mathbf{x}}(\mathbf{u}_{\mathbf{x}}, \mathbf{z}_1, \mathbf{z}_{2,e'})) = h_{\mathcal{E}}^*(\boldsymbol{x}'), \tag{4}$$

where the second equality is by Lemma 1 and the others are by definition of a spurious counterfactual. $\qquad\square$

*Proof of Lemma 1.* We step through a few key steps of the proof.

**Handling non-injective $g_{\mathbf{x}}$** First, we show that if $g_{\mathbf{x}}$ is non-injective w.r.t. the support of the original distribution $\mathbb{P}_e$, it can be written as an injective function of the support of another distribution $\tilde{\mathbb{P}}_e$, which will have the same objective value as when using $\mathbb{P}_e$. First, let $g_{\mathbf{x}}^\dagger(\boldsymbol{x})$ denote one pseudo inverse of $g_{\mathbf{x}}$ (note there could be many but we only need one here). Then, we can define the new

distribution to be equal to $\mathbb{P}_e$ except for the following:

$$\tilde{\mathbb{P}}_e(\mathbf{u_x}, \mathbf{z}_1, \mathbf{z}_2) = \begin{cases} \mathbb{P}_e(\mathbf{x} = \boldsymbol{x}), & \text{if } \exists \boldsymbol{x} \text{ such that } (\mathbf{u_x}, \mathbf{z}_1, \mathbf{z}_2) = g_{\mathbf{x}}^{\dagger}(\boldsymbol{x}) \\ 0, & \text{otherwise} \end{cases}. \tag{5}$$

We now show that this is equivalent to the objective using $\mathbb{P}_e$:

$$\mathbb{E}_{(\mathbf{x}, \mathbf{y}) \sim \mathbb{P}_e}[\ell(h(\mathbf{x}), \mathbf{y})] = \mathbb{E}_{\mathbb{P}_e(\mathbf{x})}[\mathbb{E}_{\mathbb{P}_e(\mathbf{y}|\mathbf{x})}[\ell(h(\mathbf{x}), \mathbf{y})]] \tag{6}$$

$$= \mathbb{E}_{\tilde{\mathbb{P}}_e(\mathbf{u_x}, \mathbf{z}_1, \mathbf{z}_2)}[\mathbb{E}_{\mathbb{P}_e(\mathbf{y}|\mathbf{x})}[\ell(h(g_{\mathbf{x}}(\mathbf{u_x}, \mathbf{z}_1, \mathbf{z}_2)), \mathbf{y})]] \tag{7}$$

$$= \mathbb{E}_{\tilde{\mathbb{P}}_e(\mathbf{x})}[\mathbb{E}_{\mathbb{P}_e(\mathbf{y}|\mathbf{x})}[\ell(h(\mathbf{x}), \mathbf{y})]] \tag{8}$$

$$= \mathbb{E}_{\tilde{\mathbb{P}}_e(\mathbf{x}, \mathbf{y})}[\ell(h(\mathbf{x}), \mathbf{y})], \tag{9}$$

where the second two equals are by LOTUS rules. Thus, the rest of the proof can assume that $g_{\mathbf{x}}$ is injective w.r.t. the support of $\tilde{\mathbb{P}}_e$.

**Decomposition into independent optimization problems**    Second, we show that the global min-max optimization problem can be decomposed into local min-max problems given a particular $(\mathbf{u_x}, \mathbf{z}_1)$:

$$\min_h \max_{e \in \mathcal{E}} \mathbb{E}_{(\mathbf{x}, \mathbf{y}) \sim \tilde{\mathbb{P}}_e}[\ell(h(\mathbf{x}), \mathbf{y})] \tag{10}$$

$$= \min_h \max_{f_e : e \in \mathcal{E}} \mathbb{E}_{\mathbf{u_x}, \mathbf{u_y}, \mathbf{z}_1, \mathbf{u}_2}[\ell(h(g_{\mathbf{x}}(\mathbf{u_x}, \mathbf{z}_1, f_e(\mathbf{u}_2, \mathbf{z}_1))), g_{\mathbf{y}}(\mathbf{u_y}, \mathbf{z}_1))] \tag{11}$$

$$= \min_k \max_{f_e : e \in \mathcal{E}} \mathbb{E}_{\mathbf{u_x}, \mathbf{u_y}, \mathbf{z}_1, \mathbf{u}_2}[\ell(k(\mathbf{u_x}, \mathbf{z}_1, f_e(\mathbf{u}_2, \mathbf{z}_1)), g_{\mathbf{y}}(\mathbf{u_y}, \mathbf{z}_1))] \tag{12}$$

$$= \mathbb{E}_{\mathbf{u_x}, \mathbf{z}_1}\left[ \min_{k(\mathbf{z}_2|\mathbf{u_x}, \mathbf{z}_1)} \max_{f_e(\mathbf{u}_2|\mathbf{z}_1)} \mathbb{E}_{\mathbf{u_y}, \mathbf{u}_2|\mathbf{u_x}, \mathbf{z}_1}[\ell(k(\mathbf{u_x}, \mathbf{z}_1, f_e(\mathbf{u}_2, \mathbf{z}_1)), g_{\mathbf{y}}(\mathbf{u_y}, \mathbf{z}_1))] \right], \tag{13}$$

the last step is because $k$ is injective due to $g_x$ being injective on $\tilde{p}$ so we can freely and independently choose predictions for each value of $\mathbf{u_x}$ and $\mathbf{z}_1$ in the support of $\tilde{p}$. Similarly, $f_e$ can independently and freely choose values for each value of $\mathbf{z}_1$ (it is already constant w.r.t. $\mathbf{u_x}$).

**Proving minimax solutions to independent problems are constant w.r.t. $\mathbf{z}_2$**    We will now suppress notation on $\mathbf{u_x}$ and $\mathbf{z}_1$ and simply denote $k(\mathbf{z}_2)$ and $f(\mathbf{u}_2)$. Furthermore, we will denote $\phi(\alpha) := \mathbb{E}_{\mathbf{u_y}, \mathbf{u}_2|\mathbf{u_x}, \mathbf{z}_1}[\ell(\alpha, g_{\mathbf{y}}(\mathbf{u_y}, \mathbf{z}_1))]$. Given this simplified notation, we will show that for each of these subproblems, the optimal solution to the following problem for $k$ is constant w.r.t. $\mathbf{z}_2 = f_e(\mathbf{u}_2)$:

$$\min_k \max_{f_e} \mathbb{E}_{\mathbf{u}_2}[\phi(k(f_e(\mathbf{u}_2)))]. \tag{14}$$

*Environment strategy:* The environment's optimal strategy is to concentrate all the mass on the worst case prediction:

$$\max_{f_e} \mathbb{E}_{\mathbf{u}_2}[\phi(k(f_e(\mathbf{u}_2)))] = \sup_{\boldsymbol{z}_2} \phi(k(\boldsymbol{z}_2)). \tag{15}$$

The proof of this can be seen by contradiction. If $f_e$ was optimal but varied w.r.t. $\mathbf{u}_2$, then there exists at least two measurable subsets that have different outputs. We could construct another predictor $f_e'$ by changing all the predictions to the supremum which would increase the objective, which leads to a contradiction.

*Predictor's strategy:* Now we can analyze the predictor's optimal strategy given this simplification and show that the optimal strategy is constant w.r.t. $\boldsymbol{z}_2$:

$$k^*(\boldsymbol{z}_2) := \arg\min_k \sup_{\boldsymbol{z}_2} \phi(k(\boldsymbol{z}_2)) = \arg\min_k J(k) = c. \tag{16}$$

Again, the proof is by contradiction. Suppose there was a $k$ that was optimal but was not constant. Then, we can construct a new $k'$ that will have a strictly better value: $k'(\boldsymbol{z}_2) := c = \arg\min_\alpha \phi(\alpha)$ that is a constant, where $\alpha$ is optimized over the set of possible outputs of $k$. Because there must exist at least two distinct outputs of $k$ by assumption, then we can analyze the relation between the objectives achieved for $k$ and $k'$:

$$J(k) = \sup_{\boldsymbol{z}_2} \phi(k(\boldsymbol{z}_2)) = \sup_\alpha \phi(\alpha) > \min_\alpha \phi(\alpha) = \phi(c) = \sup_{\boldsymbol{z}_2} \phi(c) = \sup_{\boldsymbol{z}_2} \phi(k'(\boldsymbol{z}_2)) = J(k'), \tag{17}$$

where the strict inequality is because $k$ was assumed to be non-constant. This leads to the contradiction that $k$ was optimal for the minimax problem. Thus, $k^*$ must be a constant w.r.t. $z_2$ as stated before. This completes the proof when combining over all values of $\mathbf{u_x}$ and $z_1$ in the support of $\tilde{\mathbb{P}}$.

The optimally robust classifier $h_{\mathcal{E}}^*$ is defined as the solution to:

$$h_{\mathcal{E}}^* = \arg\min_h J(h) = \arg\min_h \max_{e \in \mathcal{E}} \mathbb{E}_{(\mathbf{x},\mathbf{y}) \sim P_e}[\ell(h(\boldsymbol{x}), \boldsymbol{y})],$$

where $\ell$ is a strictly convex loss function. Using latent variables, and letting $k(\boldsymbol{u}_x, \boldsymbol{z}_1, \boldsymbol{z}_{2,e}) := h(g_x(\boldsymbol{u}_x, \boldsymbol{z}_1, \boldsymbol{z}_{2,e}))$, the objective function can be expressed as:

$$J(h) = \mathbb{E}_{\mathbf{u}_x, \mathbf{z}_1}\left[\max_{e \in \mathcal{E}} \mathbb{E}_{\mathbf{u}_y}\left[\mathbb{E}_{\mathbf{z}_{2,e} \sim P_e(\cdot|\boldsymbol{u}_x, \boldsymbol{z}_1)}[\ell(k(\boldsymbol{u}_x, \boldsymbol{z}_1, \boldsymbol{z}_{2,e}), g_y(\boldsymbol{u}_y, \boldsymbol{z}_1))]\right]\right].$$

The outer expectation $\mathbb{E}_{\mathbf{u}_x, \mathbf{z}_1}$ is taken because the random variables $\mathbf{u}_x$ and $\mathbf{z}_1$ (ancestors of y, $\mathbf{z}_1 = \mathbf{z}_{Anc(\mathbf{y})}$) are not affected by the choice of environment $e \in \mathcal{E}$. The term $\mathbf{z}_{2,e}$ denotes latent random variables that are non-ancestors of y, whose causal mechanisms $f_{e,i}$ (and thus their distribution $P_e(\mathbf{z}_{2,e}|\boldsymbol{u}_x, \boldsymbol{z}_1)$ conditioned on realizations $\boldsymbol{u}_x, \boldsymbol{z}_1$) can vary with $e$. The function $g_y(\boldsymbol{u}_y, \boldsymbol{z}_1)$ (target generation from realization $\boldsymbol{u}_y, \boldsymbol{z}_1$) is also invariant across environments.

To minimize $J(h)$, we need to effectively minimize the term inside the $\mathbb{E}_{\mathbf{u}_x, \mathbf{z}_1}[\cdot]$ for each pair of realizations $(\boldsymbol{u}_x, \boldsymbol{z}_1)$ independently. Let's fix $(\boldsymbol{u}_x, \boldsymbol{z}_1)$. Define:

$$\phi_{\boldsymbol{u}_x, \boldsymbol{z}_1}(\alpha) := \mathbb{E}_{\mathbf{u}_y}[\ell(\alpha, g_y(\boldsymbol{u}_y, \boldsymbol{z}_1))].$$

Since $\ell$ is strictly convex, $\phi_{\boldsymbol{u}_x, \boldsymbol{z}_1}(\alpha)$ is also strictly convex. Let $c_{\boldsymbol{u}_x, \boldsymbol{z}_1}$ be the unique minimizer of $\phi_{\boldsymbol{u}_x, \boldsymbol{z}_1}(\alpha)$:

$$c_{\boldsymbol{u}_x, \boldsymbol{z}_1} := \arg\min_\alpha \phi_{\boldsymbol{u}_x, \boldsymbol{z}_1}(\alpha).$$

This $c_{\boldsymbol{u}_x, \boldsymbol{z}_1}$ represents the optimal prediction given realizations $(\boldsymbol{u}_x, \boldsymbol{z}_1)$, averaging out $\mathbf{u}_y$, and it is independent of the realization $\boldsymbol{z}_{2,e}$ and environment $e$.

For fixed realizations $(\boldsymbol{u}_x, \boldsymbol{z}_1)$, the problem for the predictor $h$ (which chooses $k(\boldsymbol{u}_x, \boldsymbol{z}_1, \cdot)$ as a function of $\boldsymbol{z}_{2,e}$) and the environment $e$ is to determine:

$$M(k; \boldsymbol{u}_x, \boldsymbol{z}_1) = \max_{e \in \mathcal{E}}\left[\mathbb{E}_{\mathbf{z}_{2,e} \sim P_e(\cdot|\boldsymbol{u}_x, \boldsymbol{z}_1)}[\phi_{\boldsymbol{u}_x, \boldsymbol{z}_1}(k(\boldsymbol{u}_x, \boldsymbol{z}_1, \boldsymbol{z}_{2,e}))]\right].$$

The predictor $h$ chooses its function $k(\boldsymbol{u}_x, \boldsymbol{z}_1, \cdot)$ (which maps a realization $\boldsymbol{z}_{2,e}$ to a prediction value) to minimize $M(k; \boldsymbol{u}_x, \boldsymbol{z}_1)$.

1. **Environment's Strategy:** For any function $k(\boldsymbol{u}_x, \boldsymbol{z}_1, \cdot)$ chosen by $h$, the environment $e$ will choose the distribution $P_e(\mathbf{z}_{2,e}|\boldsymbol{u}_x, \boldsymbol{z}_1)$ to maximize $\mathbb{E}_{\mathbf{z}_{2,e}}[\phi_{\boldsymbol{u}_x, \boldsymbol{z}_1}(k(\boldsymbol{u}_x, \boldsymbol{z}_1, \boldsymbol{z}_{2,e}))]$. Assuming the class of SCMs $\mathcal{M}_{\mathcal{E}}$ allows the environment to concentrate probability mass, this maximum will be:

   $$\sup_{\boldsymbol{z}_{2,e}'} \phi_{\boldsymbol{u}_x, \boldsymbol{z}_1}(k(\boldsymbol{u}_x, \boldsymbol{z}_1, \boldsymbol{z}_{2,e}')).$$

   This is because the environment can choose $P_e(\mathbf{z}_{2,e}|\boldsymbol{u}_x, \boldsymbol{z}_1)$ to be a point mass (or a sequence of distributions approaching a point mass) at the realization $\boldsymbol{z}_{2,e}'$ that yields the highest value for $\phi_{\boldsymbol{u}_x, \boldsymbol{z}_1}(k(\boldsymbol{u}_x, \boldsymbol{z}_1, \boldsymbol{z}_{2,e}'))$.

2. **Predictor's Optimal Strategy (Construction):** The predictor $h$ must choose its function $k(\boldsymbol{u}_x, \boldsymbol{z}_1, \cdot)$ to minimize this supremum value:

   $$\min_{k(\boldsymbol{u}_x, \boldsymbol{z}_1, \cdot)}\left(\sup_{\boldsymbol{z}_{2,e}'} \phi_{\boldsymbol{u}_x, \boldsymbol{z}_1}(k(\boldsymbol{u}_x, \boldsymbol{z}_1, \boldsymbol{z}_{2,e}'))\right).$$

   To minimize the supremum (i.e., the worst-case value over realizations $\boldsymbol{z}_{2,e}'$) of $\phi_{\boldsymbol{u}_x, \boldsymbol{z}_1}(k(\boldsymbol{u}_x, \boldsymbol{z}_1, \boldsymbol{z}_{2,e}'))$, the optimal strategy for $k(\boldsymbol{u}_x, \boldsymbol{z}_1, \cdot)$ is to make its output constant with respect to $\boldsymbol{z}_{2,e}'$. Let this constant be $C_{\boldsymbol{u}_x, \boldsymbol{z}_1}$. Then the expression becomes $\phi_{\boldsymbol{u}_x, \boldsymbol{z}_1}(C_{\boldsymbol{u}_x, \boldsymbol{z}_1})$. The predictor will then choose this constant $C_{\boldsymbol{u}_x, \boldsymbol{z}_1}$ to be $c_{\boldsymbol{u}_x, \boldsymbol{z}_1} = \arg\min_\alpha \phi_{\boldsymbol{u}_x, \boldsymbol{z}_1}(\alpha)$, because this value minimizes $\phi_{\boldsymbol{u}_x, \boldsymbol{z}_1}(\cdot)$.

   Thus, the constructed optimal strategy for $k(\boldsymbol{u}_x, \boldsymbol{z}_1, \cdot)$ is $k(\boldsymbol{u}_x, \boldsymbol{z}_1, \boldsymbol{z}_{2,e}) = c_{\boldsymbol{u}_x, \boldsymbol{z}_1}$ for all realizations $\boldsymbol{z}_{2,e}$.

3. **Value Achieved by the Optimal Strategy:** With $k(\boldsymbol{u}_x, \boldsymbol{z}_1, \boldsymbol{z}_{2,e}) = c_{\boldsymbol{u}_x, \boldsymbol{z}_1}$, the value $M(k; \boldsymbol{u}_x, \boldsymbol{z}_1)$ becomes $\phi_{\boldsymbol{u}_x, \boldsymbol{z}_1}(c_{\boldsymbol{u}_x, \boldsymbol{z}_1})$. If $k(\boldsymbol{u}_x, \boldsymbol{z}_1, \boldsymbol{z}_{2,e})$ were any other function (i.e., not constant and equal to $c_{\boldsymbol{u}_x, \boldsymbol{z}_1}$ for all realizations $\boldsymbol{z}_{2,e}$), then there would exist some realization $\boldsymbol{z}''_{2,e}$ such that $k(\boldsymbol{u}_x, \boldsymbol{z}_1, \boldsymbol{z}''_{2,e}) \neq c_{\boldsymbol{u}_x, \boldsymbol{z}_1}$. Let $v'' = k(\boldsymbol{u}_x, \boldsymbol{z}_1, \boldsymbol{z}''_{2,e})$. Then $\phi_{\boldsymbol{u}_x, \boldsymbol{z}_1}(v'') > \phi_{\boldsymbol{u}_x, \boldsymbol{z}_1}(c_{\boldsymbol{u}_x, \boldsymbol{z}_1})$ due to the strict convexity of $\phi_{\boldsymbol{u}_x, \boldsymbol{z}_1}$ and $c_{\boldsymbol{u}_x, \boldsymbol{z}_1}$ being its unique minimizer. The adversarial environment would ensure that $\sup_{\boldsymbol{z}'_{2,e}} \phi_{\boldsymbol{u}_x, \boldsymbol{z}_1}(k(\boldsymbol{u}_x, \boldsymbol{z}_1, \boldsymbol{z}'_{2,e})) \geq \phi_{\boldsymbol{u}_x, \boldsymbol{z}_1}(v'') > \phi_{\boldsymbol{u}_x, \boldsymbol{z}_1}(c_{\boldsymbol{u}_x, \boldsymbol{z}_1})$. Thus, any strategy other than $k(\boldsymbol{u}_x, \boldsymbol{z}_1, \boldsymbol{z}_{2,e}) = c_{\boldsymbol{u}_x, \boldsymbol{z}_1}$ results in a strictly larger value for $M(k; \boldsymbol{u}_x, \boldsymbol{z}_1)$.

The overall objective $J(h)$ is $\mathbb{E}_{\mathbf{u}_x, \boldsymbol{z}_1}[M(k; \boldsymbol{u}_x, \boldsymbol{z}_1)]$. Since the optimal strategy for each pair of realizations $(\boldsymbol{u}_x, \boldsymbol{z}_1)$ is to set $k(\boldsymbol{u}_x, \boldsymbol{z}_1, \boldsymbol{z}_{2,e}) = c_{\boldsymbol{u}_x, \boldsymbol{z}_1}$, the optimally robust predictor $h^*_{\mathcal{E}}$ must be such that its corresponding $k$-function implements this strategy. Therefore, by construction of the optimal strategy for the minimax problem, it must be that:

$$h^*_{\mathcal{E}}(g_x(\boldsymbol{u}_x, \boldsymbol{z}_1, \boldsymbol{z}_{2,e})) = c_{\boldsymbol{u}_x, \boldsymbol{z}_1} \quad \text{almost everywhere.}$$

This shows that $h^*_{\mathcal{E}}(g_x(\boldsymbol{u}_x, \boldsymbol{z}_1, \boldsymbol{z}_{2,e}))$ is constant with respect to the realization $\boldsymbol{z}_{2,e}$ and equal to $c_{\boldsymbol{u}_x, \boldsymbol{z}_1}$. $\square$

## C.2 PERTURBATION THEORY

In this subsection, we revisit some important notions in the matrix perturbation theory. Let $\Delta_{\mathbf{x}}$ and $\tilde{\Delta}_{\mathbf{x}} = \Delta_{\mathbf{x}} + \varepsilon$ be two matrices in $\mathbb{R}^{d \times k}$, without loss of generality, assume $d \geq k$, as $d$ denotes the dimension of $\mathbf{x}$ and $k$ denote the counterfactual pair. Their SVDs are given respectively as follows.

$$\Delta_{\mathbf{x}} = \sum_{i=1}^{k} \sigma_i q_i (v_i)^\top = [Q_j \quad Q_{\perp,j}] \begin{bmatrix} \Sigma & 0 \\ 0 & \Sigma_{\perp} \\ 0 & 0 \end{bmatrix} \begin{bmatrix} V_j^\top \\ V_{\perp,j}^\top \end{bmatrix},$$

$$\tilde{\Delta}_{\mathbf{x}} = \sum_{i=1}^{k} \tilde{\sigma}_i \tilde{q}_i \tilde{v}_i^\top = \begin{bmatrix} \tilde{Q}_j & \tilde{Q}_{\perp,j} \end{bmatrix} \begin{bmatrix} \tilde{\Sigma} & 0 \\ 0 & \tilde{\Sigma}_{\perp} \\ 0 & 0 \end{bmatrix} \begin{bmatrix} \tilde{V}_j^\top \\ \tilde{V}_{\perp,j}^\top \end{bmatrix}.$$

Here, $\tilde{\sigma}_1 \geq \cdots \geq \tilde{\sigma}_k$ (resp. $\sigma_1 \geq \cdots \geq \sigma_k$) are the singular values of $\tilde{\Delta}_{\mathbf{x}}$ (resp. $\Delta_{\mathbf{x}}$) in descending order. $\tilde{u}_i$ (resp. $u_i$) is the left singular vector corresponding to $\tilde{\sigma}_i$ (resp. $\sigma_i$), and $v_i$ (resp. $v_i$) is the right singular vector. Define:

$$\tilde{\Sigma} := \text{diag}([\tilde{\sigma}_1, \ldots, \tilde{\sigma}_j]), \quad \tilde{\Sigma}_{\perp} := \text{diag}([\tilde{\sigma}_{j+1}, \ldots, \tilde{\sigma}_k]), \quad (18)$$

$$\tilde{Q}_j := [\tilde{q}_1, \ldots, \tilde{q}_j] \in \mathbb{R}^{d \times j}, \quad \tilde{Q}_{\perp} := [\tilde{q}_{j+1}, \ldots, \tilde{q}_d] \in \mathbb{R}^{d \times (d-j)},$$

$$\tilde{V} := [\tilde{v}_1, \ldots, \tilde{v}_j] \in \mathbb{R}^{k \times j}, \quad \tilde{V}_{\perp} := [\tilde{v}_{j+1}, \ldots, \tilde{v}_k] \in \mathbb{R}^{k \times (k-j)}.$$

The matrices $\Sigma, \Sigma_{\perp}, Q, Q_{\perp}, V, V_{\perp}$ are defined analogously.

Wedin (1972) developed a perturbation bound for singular subspaces that parallels the Davis-Kahan $\sin \Theta$ theorem for eigenspaces. The Lemma below provides bounds on the perturbation of the left and right singular subspaces.

**Lemma 2** (Wedin's $\sin \Theta$ theorem)**.** *Consider the setup in Appendix C.2. If $\sigma_1(\varepsilon) < (1 - 1/\sqrt{2})(\sigma_j - \sigma_{j+1})$, then*

$$\max \left\{ \text{dist}(\tilde{Q}_j, Q_j), \text{dist}(\tilde{V}_j, V_j) \right\} \leq \frac{2\|\varepsilon\|}{\sigma_j - \sigma_{j+1}},$$

$$\max \left\{ \text{dist}_F(\tilde{Q}_j, Q_j), \text{dist}_F(\tilde{V}_j, V_j) \right\} \leq \frac{2\sqrt{j}\|\varepsilon\|}{\sigma_j - \sigma_{j+1}}.$$

## C.3 PROOF OF THEOREM 1

We will prove the result for linear regression as it is the simplest to understand. Then, we will prove for logistic regression. And finally, we will prove the extra bound on the spurious misalignment term.

*Proof for linear regression part of Theorem 1.* Notice that by Assumption 1, we have $\mathbf{y}_e = \mathbf{y}_{e'}, \forall e, e'$, i.e., the environment does not affect the target values so we can write this as y. Therefore, we first decompose the objective by inflating by the counterfactuals $\mathbf{x}_{e^+ \to e}$:

$$
\begin{aligned}
&\mathbb{E}_{p(\mathbf{x}_{e^+})}[\ell(\theta^\top \mathbf{x}_{e^+}, \mathbf{y}_{e^+})]\\
&= \sum_{e \in \mathcal{E}_{\text{train}}} \mathbb{P}(e) \mathbb{E}_{p(\mathbf{x}_{e^+}, \mathbf{x}_{e^+ \to e})}[\|\theta^\top(\mathbf{x}_{e^+} + \mathbf{x}_{e^+ \to e} - \mathbf{x}_{e^+ \to e}) - \mathbf{y}\|_2^2]\\
&= \sum_{e \in \mathcal{E}_{\text{train}}} \mathbb{P}(e) \mathbb{E}_u[\|\theta^\top(\mathbf{x}_{e^+} + \mathbf{x}_{e^+ \to e} - \mathbf{x}_{e^+ \to e}) - \mathbf{y}\|_2^2]\\
&\leq \sum_{e \in \mathcal{E}_{\text{train}}} \mathbb{P}(e)(2\mathbb{E}_u[\|\theta^\top(\mathbf{x}_{e^+} - \mathbf{x}_{e^+ \to e})\|_2^2 + 2\mathbb{E}_u[\|\theta^\top \mathbf{x}_{e^+ \to e} - \mathbf{y}\|_2^2])\\
&= 2\sum_{e \in \mathcal{E}_{\text{train}}} \mathbb{P}(e)\mathbb{E}_u[\|\theta^\top(\mathbf{x}_{e^+} - \mathbf{x}_{e^+ \to e})\|_2^2 + 2\sum_{e \in \mathcal{E}_{\text{train}}} \mathbb{P}(e)\mathbb{E}_{p(\mathbf{x}_e, \mathbf{y})}[\|\theta^\top \mathbf{x}_e - \mathbf{y}\|_2^2]\\
&= 2\sum_{e \in \mathcal{E}_{\text{train}}} \mathbb{P}(e)\mathbb{E}_u[\|\theta^\top(\mathbf{x}_{e^+} - \mathbf{x}_{e^+ \to e})\|_2^2 + 2\mathbb{E}_{(\mathbf{x}, \mathbf{y}) \sim \mathbb{P}_{\text{train}}}[\|\theta^\top \mathbf{x} - \mathbf{y}\|_2^2],
\end{aligned}
$$

where $\mathbf{x}_{e^+ \to e}$ follows the training domain distribution $p(\mathbf{x}_e)$ and $(\mathbf{x}_{e^+}, \mathbf{x}_{e^+ \to e})$ is a conceptual counterfactual pair. Notice the second term is the training loss. Furthermore, we note that:

$$
\begin{aligned}
&\sum_{e \in \mathcal{E}_{\text{train}}} \mathbb{P}(e)\mathbb{E}_{\boldsymbol{u}}[\|\theta^\top(\mathbf{x}_{e^+} - \mathbf{x}_{e^+ \to e})\|_2^2]\\
&= \theta^\top \sum_{e \in \mathcal{E}_{\text{train}}} \mathbb{P}(e)\mathbb{E}_{\boldsymbol{u}}[(\mathbf{x}_{e^+} - \mathbf{x}_{e^+ \to e})(\mathbf{x}_{e^+} - \mathbf{x}_{e^+ \to e})^\top]\theta\\
&= \theta^\top M_{e^+, \mathcal{E}_{\text{train}}} \theta.
\end{aligned}
$$

Given this, we can simplify this term based on the NCM constraint as follows:

$$
\begin{aligned}
\theta^\top M_{e^+, \mathcal{E}_{\text{train}}} \theta &\overset{(2)}{=} \theta^\top(I - \tilde{Q}_r\tilde{Q}_r^\top)M_{e^+, \mathcal{E}_{\text{train}}}(I - \tilde{Q}_r\tilde{Q}_r^\top)^\top \theta\\
&\overset{(a)}{=} \theta^\top(I - \tilde{Q}_r\tilde{Q}_r^\top)Q_{|\mathcal{I}(\mathcal{F}_\mathcal{E})|}\Lambda_{|\mathcal{I}(\mathcal{F}_\mathcal{E})|}Q_{|\mathcal{I}(\mathcal{F}_\mathcal{E})|}^\top(I - \tilde{Q}_r\tilde{Q}_r^\top)^\top \theta\\
&= \left\|\theta^\top \underbrace{\tilde{Q}_{r,\perp}}_{\mathbb{R}^{d \times (d-r)}} \underbrace{\tilde{Q}_{r,\perp}^\top}_{\mathbb{R}^{(d-r) \times d}} \underbrace{Q_{|\mathcal{I}(\mathcal{F}_\mathcal{E})|}}_{\mathbb{R}^{d \times |\mathcal{I}(\mathcal{F}_\mathcal{E})|}} \sqrt{\Lambda_{|\mathcal{I}(\mathcal{F}_\mathcal{E})|}}\right\|^2\\
&= \left\|\theta^\top \tilde{Q}_{r,\perp}\tilde{Q}_{r,\perp}^\top Q_{|\mathcal{I}(\mathcal{F}_\mathcal{E})|}\sqrt{\Lambda_{|\mathcal{I}(\mathcal{F}_\mathcal{E})|}}\right\|^2\\
&\overset{(b)}{=} \left\|\theta^\top \tilde{Q}_{r,\perp}\tilde{Q}_{r,\perp}^\top Q_{|\mathcal{I}(\mathcal{F}_\mathcal{E})|}\right\|_{\Lambda_{|\mathcal{I}(\mathcal{F}_\mathcal{E})|}}^2\\
&= \|\theta\|^2 \left\|\tilde{Q}_{r,\perp}^\top Q_{|\mathcal{I}(\mathcal{F}_\mathcal{E})|}\right\|_{\Lambda_{|\mathcal{I}(\mathcal{F}_\mathcal{E})|}}^2,
\end{aligned}
$$

where in (a), we use eigendecomposition of $M_{e^+, \mathcal{E}_{\text{train}}}$ which $Q_{|\mathcal{I}(\mathcal{F}_\mathcal{E})|} \in \mathbb{R}^{d \times |\mathcal{I}(\mathcal{F}_\mathcal{E})|}$, and $\Lambda_{|\mathcal{I}(\mathcal{F}_\mathcal{E})|} \in \mathbb{R}^{|\mathcal{I}(\mathcal{F}_\mathcal{E})| \times |\mathcal{I}(\mathcal{F}_\mathcal{E})|}$ are corresponding eigenvectors and eigenvalues diagonal matrix and in (b), we used the definition of Mahalanobis-induced spectral norm. □

*Proof for logistic regression part of Theorem 1.* For logistic regression where $\mathbf{y} \in \{-1, 1\}$, we have a similar decomposition as in linear regression. First, we note that the log loss has a Lipschitz constant of 1 and thus we have the following:

$$
|\ell(\theta^\top \mathbf{x}, \mathbf{y}) - \ell(\theta^\top \mathbf{x}', \mathbf{y})| \leq \|\theta^\top \mathbf{x} - \theta^\top \mathbf{x}'\| = \|\theta^\top(\mathbf{x} - \mathbf{x}')\| \tag{19}
$$

$$
\Leftrightarrow \ell(\theta^\top \mathbf{x}, \mathbf{y}) \leq \ell(\theta^\top \mathbf{x}', \mathbf{y}) + \|\theta^\top(\mathbf{x} - \mathbf{x}')\|. \tag{20}
$$

Given this we can easily get our initial result:

$$
\mathbb{E}_{\mathbf{x}_{e^+}, \mathbf{y}_{e^+}}[\ell(\theta^\top \mathbf{x}_{e^+}, \mathbf{y}_{e^+})] \tag{21}
$$

$$
\leq \sum_{e \in \mathcal{E}_{\text{train}}} \mathbb{P}(e)\mathbb{E}[\ell(\theta^\top \mathbf{x}_{e^+ \to e}, \mathbf{y}) + \|\theta^\top(\mathbf{x}_{e^+} - \mathbf{x}_{e^+ \to e})\|] \tag{22}
$$

$$
= \sum_{e \in \mathcal{E}_{\text{train}}} \mathbb{P}(e)\mathbb{E}[\ell(\theta^\top \mathbf{x}_e, \mathbf{y})] + \sum_{e \in \mathcal{E}_{\text{train}}} \mathbb{P}(e)\mathbb{E}[\|\theta^\top(\mathbf{x}_{e^+} - \mathbf{x}_{e^+ \to e})\|] \tag{23}
$$

$$
= \mathbb{E}_{(\mathbf{x}, \mathbf{y}) \sim \mathbb{P}_{\text{train}}}[\ell(\theta^\top \mathbf{x}, \mathbf{y})] + \sum_{e \in \mathcal{E}_{\text{train}}} \mathbb{P}(e)\mathbb{E}[\|\theta^\top(\mathbf{x}_{e^+} - \mathbf{x}_{e^+ \to e})\|], \tag{24}
$$

where we use the fact that $\mathbf{x}_{e^+ \to e}$ has the same distribution as $\mathbf{x}_e$. Now we bound the second term as follows:

$$\sum_{e \in \mathcal{E}_{\text{train}}} \mathbb{P}(e) \mathbb{E}[\|\theta^\top (\mathbf{x}_{e^+} - \mathbf{x}_{e^+ \to e})\|] \tag{25}$$

$$= \sum_{e \in \mathcal{E}_{\text{train}}} \mathbb{P}(e) \mathbb{E}[\sqrt{\|\theta^\top (\mathbf{x}_{e^+} - \mathbf{x}_{e^+ \to e})\|^2}] \tag{26}$$

$$\leq \sqrt{\sum_{e \in \mathcal{E}_{\text{train}}} \mathbb{P}(e) \mathbb{E}[\|\theta^\top (\mathbf{x}_{e^+} - \mathbf{x}_{e^+ \to e})\|^2]} \tag{27}$$

$$= \sqrt{\theta^\top \left( \sum_{e \in \mathcal{E}_{\text{train}}} \mathbb{P}(e) \mathbb{E}[(\mathbf{x}_{e^+} - \mathbf{x}_{e^+ \to e})(\mathbf{x}_{e^+} - \mathbf{x}_{e^+ \to e})^\top] \right) \theta} \tag{28}$$

$$= \sqrt{\theta^\top M_{e^+, \mathcal{E}_{\text{train}}} \theta} \tag{29}$$

$$\leq \sqrt{\|\theta\|^2 \left\| \tilde{Q}_{r,\perp}^\top Q_{|\mathcal{I}(\mathcal{F}_\mathcal{E})|} \right\|_{\Lambda_{|\mathcal{I}(\mathcal{F}_\mathcal{E})|}}^2} \tag{30}$$

$$= \|\theta\| \left\| \tilde{Q}_{r,\perp}^\top Q_{|\mathcal{I}(\mathcal{F}_\mathcal{E})|} \right\|_{\Lambda_{|\mathcal{I}(\mathcal{F}_\mathcal{E})|}}, \tag{31}$$

where (27) is by Jensen's inequality and (30) is by using the same logic as in the linear regression case for this term. Combining the above derivations, we get the following:

$$\mathbb{E}_{\mathbf{x}_{e^+}, \mathbf{y}_{e^+}} [\ell(\theta^\top \mathbf{x}_{e^+}, \mathbf{y}_{e^+})] \tag{32}$$

$$\leq \mathbb{E}_{(\mathbf{x}, \mathbf{y}) \sim \mathbb{P}_{\text{train}}} [\ell(\theta^\top \mathbf{x}, \mathbf{y})] + \sum_{e \in \mathcal{E}_{\text{train}}} \mathbb{P}(e) \mathbb{E}[\|\theta^\top (\mathbf{x}_{e^+} - \mathbf{x}_{e^+ \to e})\|] \tag{33}$$

$$\leq \mathbb{E}_{(\mathbf{x}, \mathbf{y}) \sim \mathbb{P}_{\text{train}}} [\ell(\theta^\top \mathbf{x}, \mathbf{y})] + \sqrt{\|\theta\|^2 \left\| \tilde{Q}_{r,\perp}^\top Q_{|\mathcal{I}(\mathcal{F}_\mathcal{E})|} \right\|_{\Lambda_{|\mathcal{I}(\mathcal{F}_\mathcal{E})|}}^2} \tag{34}$$

$$= \mathbb{E}_{(\mathbf{x}, \mathbf{y}) \sim \mathbb{P}_{\text{train}}} [\ell(\theta^\top \mathbf{x}, \mathbf{y})] + \|\theta\| \left\| \tilde{Q}_{r,\perp}^\top Q_{|\mathcal{I}(\mathcal{F}_\mathcal{E})|} \right\|_{\Lambda_{|\mathcal{I}(\mathcal{F}_\mathcal{E})|}}. \tag{35}$$

$$\square$$

*Proof of bound on the orthonormal term.* In this proof, we seek to prove the following:

$$\left\| \tilde{Q}_{r,\perp}^\top Q_{|\mathcal{I}(\mathcal{F}_\mathcal{E})|} \right\|_{\Lambda_{|\mathcal{I}(\mathcal{F}_\mathcal{E})|}}^2 \leq \lambda_1(e^+) \text{dist}^2(\tilde{Q}_s, Q_s) + \lambda_{s+1}(e^+). \tag{36}$$

We will consider two cases to bound the result depending on whether $r$ is greater or less than $|\mathcal{I}(\mathcal{F}_\mathcal{E})|$.

**Case I**: $r < |\mathcal{I}(\mathcal{F}_\mathcal{E})|$. Given this case, we can decompose as follows:

$$\left\| \tilde{Q}_{r,\perp}^\top Q_{|\mathcal{I}(\mathcal{F}_\mathcal{E})|} \right\|_{\Lambda_{|\mathcal{I}(\mathcal{F}_\mathcal{E})|}}^2 = \left\| \tilde{Q}_{r,\perp}^\top Q_{|\mathcal{I}(\mathcal{F}_\mathcal{E})|} \Lambda_{|\mathcal{I}(\mathcal{F}_\mathcal{E})|}^{\frac{1}{2}} \right\|^2 \tag{37}$$

$$= \left\| \tilde{Q}_{r,\perp}^\top Q_r \Lambda_r^{\frac{1}{2}} \right\|^2 + \left\| \tilde{Q}_{r,\perp}^\top Q_{r+1:|\mathcal{I}(\mathcal{F}_\mathcal{E})|} \Lambda_{r+1:|\mathcal{I}(\mathcal{F}_\mathcal{E})|}^{\frac{1}{2}} \right\|^2 \tag{38}$$

$$\leq \lambda_1(e^+) \left\| \tilde{Q}_{r,\perp}^\top Q_r \right\|^2 + \lambda_{r+1}(e^+) \left\| \tilde{Q}_{r,\perp}^\top Q_{r+1:|\mathcal{I}(\mathcal{F}_\mathcal{E})|} \right\|^2 \tag{39}$$

$$\leq \lambda_1(e^+) \left\| \tilde{Q}_{r,\perp}^\top Q_r \right\|^2 + \lambda_{r+1}(e^+) \tag{40}$$

$$\leq \lambda_1(e^+) \left\| \tilde{Q}_r \tilde{Q}_r^\top - Q_r Q_r^\top \right\|^2 + \lambda_{r+1}(e^+) \tag{41}$$

$$= \lambda_1(e^+) \text{dist}^2(\tilde{Q}_r, Q_r) + \lambda_{r+1}(e^+), \tag{42}$$

where (39) is by the properties of norms, (40) is by the fact that $\|QQ'\|$ for any orthogonal matrices is always less than 1, (41) is from Chen et al. (2021), and (42) is by the definition of the dist function between two subspaces.

**Case II**: $r \geq |\mathcal{I}(\mathcal{F}_\mathcal{E})|$. Given this condition, we can get a simpler case without the second term:

$$\left\| \tilde{Q}_{r,\perp}^\top Q_{|\mathcal{I}(\mathcal{F}_\mathcal{E})|} \right\|_{\Lambda_{|\mathcal{I}(\mathcal{F}_\mathcal{E})|}}^2 = \left\| \tilde{Q}_{r,\perp}^\top Q_{|\mathcal{I}(\mathcal{F}_\mathcal{E})|} \Lambda_{|\mathcal{I}(\mathcal{F}_\mathcal{E})|}^{\frac{1}{2}} \right\|^2 \tag{43}$$

$$= \left\| \tilde{Q}_{|\mathcal{I}(\mathcal{F}_\mathcal{E})|,\perp}^\top Q_{|\mathcal{I}(\mathcal{F}_\mathcal{E})|} \Lambda_{|\mathcal{I}(\mathcal{F}_\mathcal{E})|}^{\frac{1}{2}} \right\|^2 \tag{44}$$

$$\leq \lambda_1(e^+) \left\| \tilde{Q}_{|\mathcal{I}(\mathcal{F}_\mathcal{E})|,\perp}^\top Q_{|\mathcal{I}(\mathcal{F}_\mathcal{E})|} \right\|^2 \tag{45}$$

$$\leq \lambda_1(e^+) \| \tilde{Q}_{|\mathcal{I}(\mathcal{F}_\mathcal{E})|} \tilde{Q}_{|\mathcal{I}(\mathcal{F}_\mathcal{E})|}^\top - Q_{|\mathcal{I}(\mathcal{F}_\mathcal{E})|} Q_{|\mathcal{I}(\mathcal{F}_\mathcal{E})|}^\top \|^2 \tag{46}$$

$$= \lambda_1(e^+) \mathrm{dist}^2 (\tilde{Q}_{|\mathcal{I}(\mathcal{F}_\mathcal{E})|}, Q_{|\mathcal{I}(\mathcal{F}_\mathcal{E})|}), \tag{47}$$

where (44) is by the condition of Case II and the others follow similarly from Case I. Now we can combine both of these cases to form a bound based on $s := \min\{r, |\mathcal{I}(\mathcal{F}_\mathcal{E})|\}$ to yield the final result:

$$\left\| \tilde{Q}_{r,\perp}^\top Q_{|\mathcal{I}(\mathcal{F}_\mathcal{E})|} \right\|_{\Lambda_{|\mathcal{I}(\mathcal{F}_\mathcal{E})|}}^2 \leq \lambda_1(e^+) \mathrm{dist}^2(\tilde{Q}_s, Q_s) + \lambda_{s+1}(e^+) \,.$$

$\square$

## C.4 PROOF OF TEST-DOMAIN BOUNDS IN TERMS OF COUNTERFACTUAL NOISE

*Proof of Corollary 3.* First, by the rank condition on the corresponding clean counterfactuals, we know that the clean counterfactual matrix $\Delta_\mathbf{x}$ must only span the spurious subspace. Thus, the eigenvectors of $\Delta_\mathbf{x} \Delta_\mathbf{x}^\top$ (denoted by $Q_{\Delta_\mathbf{x}}$ are equivalent to the left singular vectors of $M_{e^+, \mathcal{E}_{\mathrm{train}}}$, i.e., $Q \equiv Q_{M_{e^+, \mathcal{E}_{\mathrm{train}}}} = Q_{\Delta_\mathbf{x}}$.

Based on this, we can derive the result where we let $s := \min(r, |\mathcal{I}(\mathcal{F}_\mathcal{E})|)$:

$$\mathrm{dist}(\tilde{Q}_s, Q_s) = \mathrm{dist}(\tilde{Q}_{\tilde{\Delta}_\mathbf{x}, s}, Q_{\Delta_\mathbf{x}, s}) \leq \frac{2\|\varepsilon\|}{\sigma_s - \sigma_{s+1}} \,,$$

where the first is by the definition of $\tilde{Q}$ and the fact above, and the inequality is by Wedin's theorem (Lemma 2). Combining this with the original results in Theorem 1, we arrive at the bound. $\square$

## C.5 PERFECT COUNTERFACTUAL CASE

We now consider the noiseless counterfactual case. We can easily prove a corollary that the test-domain error is equal to the train domain error if the counterfactuals are diverse enough (i.e., they satisfy the rank condition of Corollary 3).

**Corollary 4.** *Instate the setting from Corollary 3. Further, assume that $r \geq |I(\mathcal{F}_\mathcal{E})|$ and the noise is zero, i.e., $\varepsilon = 0$, then we can derive that the test-domain error for any $\theta$ satisfying the NCM constraint is equal to the training error for both logistic and linear regression:*

$$\mathbb{E}_{(\mathbf{x}_{e^+}, \mathbf{y}_{e^+}) \sim \mathbb{P}_{\mathrm{test}}} \left[ \ell(\theta^\top \mathbf{x}_{e^+}, \mathbf{y}_{e^+}) \right] = \mathbb{E}_{(\mathbf{x}, \mathbf{y}) \sim \mathbb{P}_{\mathrm{train}}} \left[ \ell(\theta^\top \mathbf{x}, \mathbf{y}) \right] \,. \tag{48}$$

*Proof.* First, we note that the inequality for squared error that introduces the 2 can be removed for perfect counterfactuals because the term is 0 and doesn't need to use bounds. Other than that, we can simply apply the result from Corollary 3 and note that $\lambda_{s+1}(e^+)$ will inherently be 0 due to the $r \geq |I(\mathcal{F}_\mathcal{E})|$ assumption and the $\|\varepsilon\| = 0$ by assumption as well. $\square$

We validate this result using the synthetic dataset in Section 5 (see Figure 2a) with $\varepsilon = 0$, showing that when $k \geq |\mathcal{I}(\mathcal{F}_\mathcal{E})|$, the model achieves optimal performance.

The following comments are in order. **1)** Simple CF pair-matching (1) provably generalizes to new test domains. If we run an algorithm $\mathcal{A}$ to solve (1), and if it can perform well on the training domain, then, it will also perform well on the test domain. **2)** A linear number of oracle counterfactual pairs is sufficient to achieve domain generalization. By assuming that the differences are linearly independent, the required number of counterfactual pairs is bounded by $k \geq |\mathcal{I}(\mathcal{F}_\mathcal{E})|$. This implies that each counterfactual pair effectively removes one spurious dimension. Even when the data dimension $d$ is high in reality, by the sparse mechanism shift hypothesis, the spurious mechanism shift

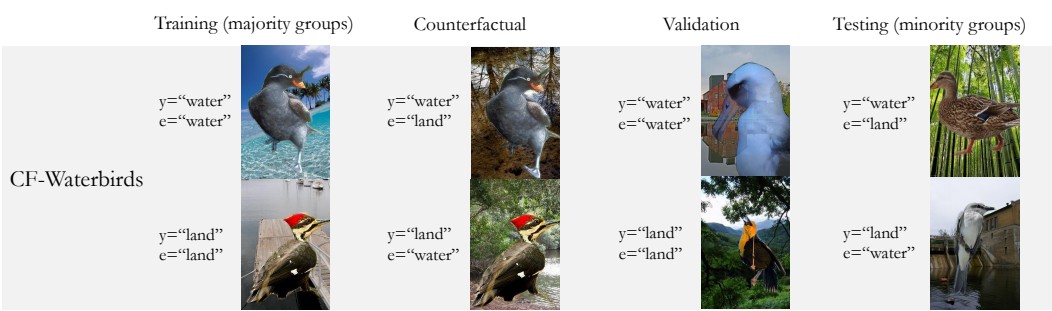

Figure 6: Illustration of the training samples, counterfactual pairs, validation examples, and test examples for the Waterbirds-CF dataset. The training set (majority groups), validation set, and test set are identical to those in the original Waterbirds dataset. Counterfactual pairs feature the same birds from the training set but with different backgrounds.

(Schölkopf et al., 2021) suggests that the spurious feature space is indeed low, thereby supporting the effectiveness of our proposed method.

A another case arises when $e^+$ chosen to be sampled from mixture of training domains. In this case, the spurious subspace misalignment vanishes as the test domain is already seen, thus $\mathbf{x}_{e^+} = \mathbf{x}_{e^+ \to e}$ i.e., $\lambda_1(e^+) = 0$, NCM objective (2) and simple CF pair-matching (1) reduce to empirical risk minimization (ERM).

## D  DETAILED EXPERIMENT SETUP AND HYPERPARAMETERS

We provide a detailed description of our experiments setup and hyperparameter selection.

### D.1  DETAILED CONSTRUCTION OF WATERBIRDS-CFS

The original Waterbirds dataset (Sagawa et al., 2019) combines bird images from the CUB dataset (Wah et al., 2011) with background images from the Places dataset (Zhou et al., 2017). The task is to classify whether a given image depicts a waterbird ($y = 1$) or a landbird ($y = 0$). Waterbirds include seabirds (albatross, auklet, cormorant, frigatebird, fulmar, gull, jaeger, kittiwake, pelican, puffin, and tern) and waterfowl (gadwall, grebe, mallard, merganser, guillemot, and Pacific loon).

In the dataset, the invariant features are represented by the bird segments, while the spurious features are the backgrounds. In the training set, the background is highly correlated with the bird species: 95% of waterbirds appear in a water background (ocean or natural lake), and similarly, 95% of landbirds are shown against a land background (bamboo or broadleaf forest). The remaining 5% consist of counterfactual samples, which are random samples from the majority group. Counterfactual pairs share the same bird segment but differ in the background. The validation and test sets are identical to the original Waterbirds dataset, meaning the conditional distribution of the background given either waterbirds or landbirds is 50%.

In summary, the modification made between our Waterbirds-CF dataset and the original waterbirds dataset only pertains to the minority groups in the training set. We randomly sampled 184 landbirds and 56 waterbirds from the majority group and replaced the backgrounds of these samples to generate the minority group counterfactuals. See Figure 6 for the illustration. We then applied ERM on both these two datasets to investigate the changing in the training distribution. We include the convergence curve in Appendix E.2.

### D.2  HYPERPARAMETER SELECTION

In this section, we present all the hyperparameters and evaluation used in our experiments to ensure reproducibility.

**Synthetic Dataset**   Invariant features are sampled from a standard normal distribution, i.e., $\mathbf{z}_{\text{inv}} \sim \mathcal{N}(0, I)$, The observation function $g_y$ is linear, with parameter $\theta_y \sim \mathcal{N}(0, \sigma I)$, and the label $y = \text{sign}(\mathbf{z}_{\text{inv}}\theta_y)$. The spurious features is correlated to the label y, i.e., $\mathbf{z}_{\text{spu}} \sim \mathcal{N}\left(\frac{y}{|\mathcal{I}(\mathcal{F}_\varepsilon)|}, \sigma_s I\right)$ where $\sigma_s$ varies across domains. The observation function $g_x$ is a random orthonormal matrix. The dimension of $\mathbf{z}$ and $\mathbf{x}$ are both 100, i.e., $m = d = 100$.

We run 100 iterations of gradient descent using binary cross-entropy loss. We use the Adam optimizer (Kingma, 2014) with a learning rate of 0.01 for ERM, IRM, and NCM. The Lagrange multiplier for both IRM and NCM is set to $\lambda = 1000$ selected through grid search.

**ColoredMNIST**   We use the Adam optimizer with a learning rate of 0.001 and a weight decay of $10^{-4}$. The model is trained with a batch size of 256 for 40 epochs. We tune the hyperparameter r in the range [2, 24] using 256 counterfactual pairs.

**Waterbirds-CF**   We use the Adam optimizer with a learning rate of 0.001 and a weight decay of $10^{-4}$. The model is trained with a batch size of 256 for 100 epochs. We tune the hyperparameter r in the range [2, 24].

**PACS**   We use the Adam optimizer with a learning rate of 0.01 and a weight decay of $10^{-4}$. The model is trained with a batch size of 256 for 100 epochs. We tune the hyperparameter r in the range [2, 24].

Details on hyperparameter tuning and baseline method selection can be found in the code repository.

# E   ADDITIONAL EXPERIMENTS

Table 3: Main Results on ColoredMNIST

| | in-domain validation | | oracle validation | |
|---|---|---|---|---|
| | in acc | test acc | in acc | test acc |
| ERM (CLIP) | 0.852 | 0.093 | 0.753 | 0.253 |
| IRM | 0.799 | 0.118 | 0.724 | 0.469 |
| REx | 0.797 | 0.121 | 0.691 | 0.664 |
| GroupDRO | 0.798 | 0.127 | 0.786 | 0.201 |
| Fish | 0.798 | 0.118 | 0.495 | 0.486 |
| SWAD | 0.800 | 0.113 | 0.501 | 0.505 |
| LISA | 0.705 | **0.693** | 0.705 | 0.693 |
| MatchDG w. random | 0.799 | 0.120 | 0.511 | 0.512 |
| MatchDG w. 1NN | 0.789 | 0.217 | 0.728 | 0.662 |
| MatchDG w. clean | 0.793 | 0.181 | 0.742 | 0.672 |
| NCM w. random | 0.794 | 0.176 | 0.680 | 0.706 |
| NCM w. 1NN | 0.736 | 0.649 | 0.711 | 0.707 |
| NCM w. clean | 0.740 | **0.693** | 0.727 | **0.714** |
| random guess | 0.500 | 0.500 | 0.500 | 0.500 |
| ERM oracle | 0.735 | 0.730 | 0.735 | 0.730 |
| theory oracle | 0.750 | 0.750 | 0.750 | 0.750 |

Table 4: Main Results on Waterbirds-CF

| | In-domain Validation | | Oracle Validation | |
|---|---|---|---|---|
| | in acc | wg acc | in acc | wg acc |
| ERM (CLIP) | 0.885 | 0.781 | 0.882 | 0.800 |
| ERM+UW | 0.889 | 0.795 | 0.882 | 0.829 |
| IRM | 0.838 | 0.707 | 0.820 | 0.767 |
| REx | 0.891 | 0.617 | 0.878 | 0.729 |
| GroupDRO | 0.906 | 0.684 | 0.896 | 0.827 |
| Fish | 0.900 | 0.744 | 0.869 | 0.805 |
| LISA | 0.904 | 0.722 | 0.876 | 0.812 |
| MatchDG w. random | 0.793 | 0.009 | 0.785 | 0.149 |
| MatchDG w. 1NN | 0.886 | 0.411 | 0.886 | 0.411 |
| MatchDG w. estimated CF | 0.906 | 0.536 | 0.896 | 0.651 |
| NCM w. random | 0.804 | 0.269 | 0.804 | 0.269 |
| NCM w. 1NN | 0.892 | 0.521 | 0.882 | 0.560 |
| NCM w. estimated CF | 0.864 | **0.812** | 0.854 | **0.860** |

In this section, we include more experiments results. We further illustrate the effectiveness of our method, as well as the hyperparameters sensitivity.

## E.1   ABLATION STUDY

**Sensitivity on truncated SVD parameter** $r$.   We empirically evaluate the trade-off effect of the hyperparameter $r$ on model performance during linear probing on the Waterbirds-CF dataset (cf. Figure 7), thus validating Theorem 1 comment **(iii)**: accuracy trade-off induced by $r$. This pattern

Table 5: Main Results on PACS

| | In-domain Validation | | | | | Oracle Validation | | | | |
| --- | --- | --- | --- | --- | --- | --- | --- | --- | --- | --- |
| | A | C | P | S | Avg | A | C | P | S | Avg |
| ERM (CLIP) | 0.924 | 0.968 | 0.996 | 0.859 | 0.937 | 0.924 | 0.968 | 0.996 | 0.859 | 0.937 |
| IRM | 0.938 | 0.976 | 0.996 | 0.840 | 0.938 | 0.941 | 0.976 | 0.996 | 0.845 | 0.940 |
| REx | 0.953 | 0.963 | 0.993 | 0.836 | 0.936 | 0.953 | 0.975 | 0.996 | 0.845 | 0.942 |
| GroupDRO | 0.903 | 0.963 | 0.996 | 0.873 | 0.934 | 0.941 | 0.975 | 0.996 | 0.843 | 0.939 |
| Fish | 0.936 | 0.973 | 0.996 | 0.837 | 0.936 | 0.936 | 0.973 | 0.996 | 0.837 | 0.936 |
| SWAD | 0.941 | 0.976 | 0.996 | 0.838 | 0.938 | 0.941 | 0.977 | 0.996 | 0.838 | 0.938 |
| LISA | 0.926 | **0.978** | 0.997 | 0.848 | 0.937 | 0.940 | **0.983** | 0.997 | 0.864 | 0.946 |
| MatchDG w. rand. | 0.412 | 0.509 | 0.316 | 0.749 | 0.497 | 0.454 | 0.509 | 0.358 | 0.749 | 0.518 |
| MatchDG w. 1NN. | **0.964** | 0.971 | 0.995 | 0.880 | **0.953** | 0.964 | 0.973 | 0.996 | **0.887** | **0.955** |
| NCM w. rand. | 0.591 | 0.609 | 0.577 | 0.833 | 0.653 | 0.592 | 0.625 | 0.583 | 0.843 | 0.661 |
| NCM w. 1NN. | 0.957 | 0.974 | **0.998** | **0.882** | **0.953** | **0.964** | 0.974 | **0.998** | 0.885 | **0.955** |

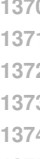
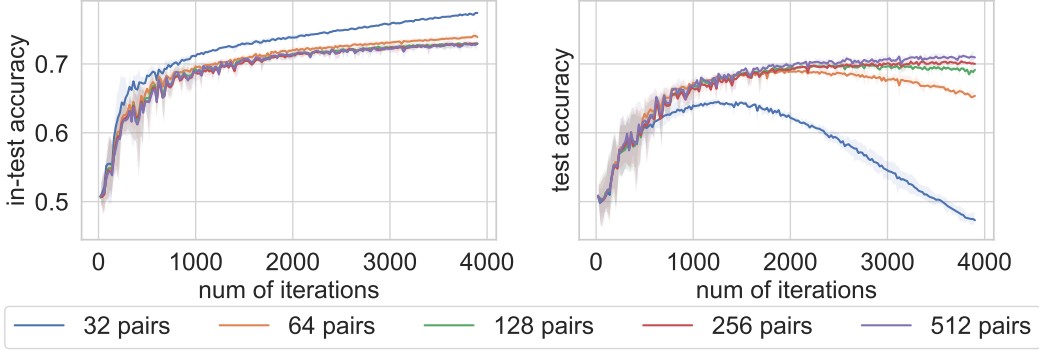

Figure 7: In-domain test and worst-group accuracy with changing hyperparameter $r$. In-domain accuracy remains stable for small values of $r$, but starts to drop at $r \approx 128$. Worst-group accuracy first increases then decreases as $r$ grows.

Figure 8: The number of counterfactuals vs. in-test accuracy curve and test accuracy curve on ColoredMNIST using the CLIP + Linear model. We conduct evaluations using 32, 64, 128, 256, and 512 data pairs.

reflects the model's shifting reliance from spurious to invariant features: when $r$ is too small, spurious correlations dominate, resulting in high in-domain but low worst-group performance. As $r$ increases and suppresses these spurious features, worst-group accuracy improves. However, beyond a certain point, further increases in $r$ begin to remove invariant features as well, leading to a decline in both metrics.

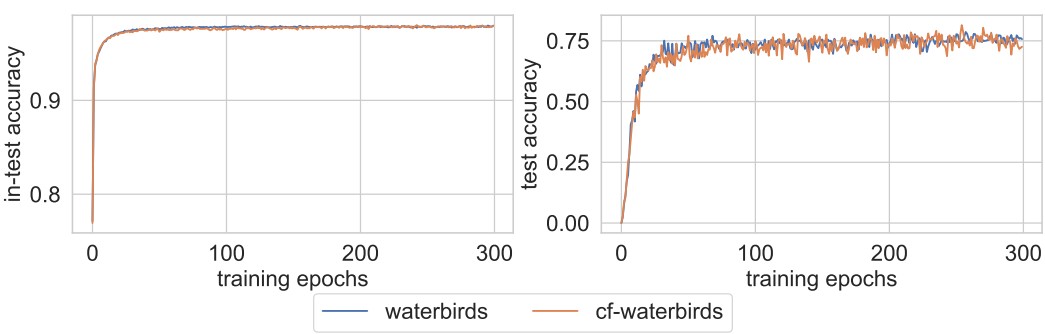

Figure 9: Compare the convergence curve of in-domain average test accuracy as well as the worst-case test accuracy on waterbirds and Waterbirds-CF datasets

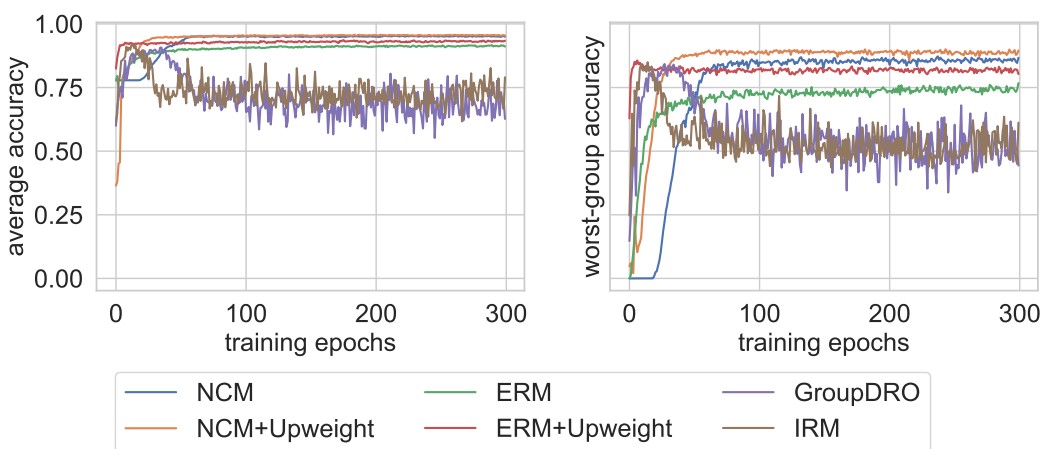

Figure 10: The convergence curve of Waterbirds-CF dataset. It shows that our method is significantly more stable than other DG methods without suffering overfitting.

**Sensitivity on the number of CF pairs.** We evaluate the number of counterfactual pairs needed on ColoredMNIST dataset and report the in-domain test accuracy and test accuracy with 32,64,128,256,512 CF pairs. The results show that with 32 counterfactual pairs, the number of pairs is insufficient for the model to eliminate spurious features, leading to spurious correlations (as indicated by an in-domain accuracy over 75%, meaning the classification relies on spurious features). However, when using 128 or 256 counterfactual pairs, the performance increases significantly and remains stable compared to the 32 counterfactual pairs. An insufficient number of pairs fails to eliminate the spurious feature, allowing the model to eventually rely on it, which leads to decreased accuracy on the test domain.

### E.2 COMPARISON BETWEEN WATERBIRDS AND WATERBIRDS-CF ON ERM.

We run ERM on both waterbirds dataset and our Waterbirds-CF dataset. The results of ERM on both datasets are almost identical (cf. Figure 9).

### E.3 BEYOND LINEARITY

Though our NCM relies on linear assumption, our method could further work under nonlinear models empirically. In this section, we consider waterbirds-cf dataset using ResNet dataset. We apply mini-batch SGD with 300 epochs on the pretrained ResNet50 (He et al., 2016)[3]. The optimizer used is SGD with a step size of $0.001$, momentum of $0.9$, and weight decay of $0.0001$, as recommended for the Waterbirds dataset. The batch size is set to 128, For each batch, 128 counterfactual pairs

---

[3]pretrained model is IMAGENET1K_V1 from torchvision. Download here.

Table 6: Waterbirds-CF results on ResNet-50: best performance over 300 epochs, averaged across 5 runs. Adjusted accuracy is the reweighted metric to match the training distribution. Avg. Acc and WG. Acc denotes average accuracy and worst-group accuracy respectively.

|  | Oracle Validation | | |
|---|---|---|---|
|  | In-domain Acc | Test Acc | Worst domain Acc |
| ERM | 0.978 | 0.917 | 0.767 |
| ERM + UW | 0.980 | 0.958 | 0.856 |
| IRM | 0.943 | 0.920 | 0.849 |
| GroupDRO | 0.934 | 0.907 | 0.842 |
| NCM w. oracle pairing | 0.978 | 0.953 | 0.872 |
| NCM w. oracle pairing +UW | 0.980 | 0.957 | 0.900 |

are sampled to form the constraint term, which these pairs are matched prior to the linear classifier with MSE loss, The latent dimensionality is $64$. The Lagrange multiplier is set to $500$ (and $100$ for IRM). For GroupDRO, we set the learning rate for updating the weight to be $0.01$. All these hyperparameters are selected through grid search. We report the convergence curve of our methods as well as comparison to other baselines in Table 4. In the table, we report all the methods' best performance on average over 300 epochs of running. From the result we show that our NCM using only 240 counterfactual pairs, outperforms ERM by 10.5% on the worst group accuracy. Further, we outperform other baselines like IRM and GroupDRO by 3.0% and 2.6%. Observe that Figure 10 shows that NCM is much more stable compared to IRM and GroupDRO. As mentioned that NCM is a causal data-centric approach, it could be combined with existed method to further improve domain generalization potentially. Here, we combine our method with up-weighting technique and we get 4.4% improvement over the ERM up-weighting counterpart. We further include experiments on the sensitivity of hyperparameters in Appendix D.

