# OpenReview forum: "From Invariant Representations to Invariant Data: Provable Robustness to Spurious Correlations via Noisy Counterfactual Matching"
_ICLR.cc/2026/Conference — Submitted to ICLR 2026_

### Official Review · Reviewer_FWNp · 2025-10-24

**Soundness:** 3
**Presentation:** 3
**Contribution:** 2
**Rating:** 2
**Confidence:** 4

**Summary:**

The paper tackles domain generalization problems with a shift in focus from learning invariant representations to leveraging invariant data, where an invariant pair is a pair of samples that should share the same prediction under the optimal hypothesis. The authors first show that counterfactual pairs are invariant and analyzes when such pairs improve robustness and how many are needed. The paper then introduces Noisy Counterfactual Matching, which adds a constraint to ERM to improve robustness to spurious correlations using a small set of (noisy) invariant pairs. The scope of distribution shift considered is restricted to domains that only intervene on spurious latent variables that are non‑ancestors of the target variable.

**Strengths:**

* The paper is clearly written and easy to follow.
* The problem is well motivated, with an interesting insight and thorough theoretical analyses for the setting considered.
* Empirical results are presented to support the theoretical claims.

**Weaknesses:**

* Theoretical results are only applicable to linear SCMs, which is rather simple and restrictive, relative to real‑world scenarios.

* The collection of invariant data pairs is only feasible given some knowledge about the spurious features. While the authors argue that such knowledge may be available in practice, I think that the knowledge of some may be feasible, but knowledge of all possible spurious features is hard to achieve, even with domain knowledge. For data like images, spurious features may simply be related to the background or style of the images. However for tabular data in arbitrary more complex domains, humans are often deprived of the full knowledge of all possible spurious factors.

* There are other technical concerns that requires further clarification. See the questions below.

**Questions:**

1. The logic of the proposal seems problematic to me. The paper proves that counterfactual pairs improve robustness and then uses the fact that counterfactual pairs are invariant to justify collecting invariant pairs instead. Are counterfactual and invariant pairs equivalent? From my understanding of its proof, Proposition 1 shows that counterfactual pairs are invariant, but not all invariant pairs are counterfactual. If the model gives the same prediction to different inputs, collecting such pairs may not yield the desired robustness effect. Could the authors clarify this?

2. The examples about the collection of invariant pairs, in the Introduction and Appendix A.2, seem to rely on knowing that pairs share the same label, whereas invariance is defined with respect to the optimal hypothesis. True labels and optimal predictions can differ. How does this mismatch affect your results? If invariant pairs can be collected using training labels, why not define invariance directly with respect to the true label and perform the alignment/subspace objective accordingly, rather than relying on the robust model’s prediction?

3. What is the output space of the robust classifier $h$? Does it output a hard label or a label distribution?

4. Definition 1 is stated for classifiers, but Theorem 1 discusses linear and logistic regression losses. Is Theorem 1 applicable to common losses like cross‑entropy for classification tasks?

5. The authors also claim that one feasible way to collect invariant pairs is to perform augmentation. So at a high level, I do not understand how the proposed approach differs from data augmentation or in other words directly using augmented data for alignment.

6. A major limitation is that the constraint in Eq. (1) reduces to Eq. (2) only under linear models. Once we have the counterfactual data pairs, why don’t we directly align them using some distance metrics, as often done in machine learning literature e.g., contrastive learning?

7. I understand that theoretical analyses in non-linear settings are challenging. However, could the authors at least provide empirical results on standard benchmark like DomainBed to at least verify whether the idea carries over empirically?

My current rating reflects the above concerns. I will consider updating the score once they are resolved.

---

> ### Author Response · Authors · 2025-11-21
> **Rebuttal (Part 1 of 3)**
>
> We thank the reviewer for their valuable time and detailed feedback. Our responses to the concerns and questions are provided below.
>
> > Theoretical results are only applicable to linear SCMs ...
>
> Given our theoretical analysis and the utility of linear probing on top of representation models in practice, we argue that our linearity assumption is not unusually restrictive. Rather, it helps us isolate core concepts and provide the foundation for non-linear analysis, which we agree is an important future research direction. We explain more below:
> 1. **Linear SCMs are a canonical theoretic testbed isolating the core challenges.**
> Linear SCMs [1-9] are a standard starting point in causal and domain-generalization theory: they allow one to isolate some other challenges to obtain sharp, interpretable results about identifiability and robustness. In our setting, the linear SCM assumption lets us precisely characterize how the spurious subspace relates to counterfactual differences and when a small set of invariant pairs suffices to recover the robust predictor. For example, ColoredMNIST might be thought of as an overly easy task in the era of LLM, it is indeed shown to be a very hard generalization task. [10]
> 2. **The results are about geometry, not just about “toy” models.**
> Our guarantees are fundamentally geometric: they relate out-of-domain error to (i) the alignment between the subspace spanned by invariant-pair differences and the true intervention subspace, and (ii) spectral properties of the corresponding covariance. This provides conceptual guidance that extends beyond strictly linear generative processes. For example, the same geometric picture underlies our use of NCM as a linear probe on top of powerful nonlinear encoders (e.g., CLIP) in the experiments.
> 3. **Theory plus practice: non-linear feature maps in experiments.**
> While the SCM analysis is linear, our empirical evaluation is explicitly non-linear: we apply NCM to CLIP features and other learned representations. In this sense, the theory clarifies why the algorithm should be effective once an approximately linear “task+spurious” structure is exposed in feature space, and the experiments demonstrate that this intuition carries over to realistic, high-dimensional settings.
> 4. **Scope and future extensions.**
> We fully acknowledge that extending the analysis to more general (e.g., nonlinear or non-additive) SCMs is an important direction for future work. Our goal in this paper is to establish a first, clean characterization of how invariant pairs control robustness under interventions, and to show that even in the linear case this already yields nontrivial, practically useful insights and algorithms.
>
> > The collection of invariant data pairs is only ...
>
> We respectfully disagree with the premise of this concern. Our goal is not to guarantee robustness against all possible spurious features in the DAG. Instead, our framework is designed for settings where practitioners have a specific robustness target in mind.
> Concretely, even if the underlying causal graph contains many spurious factors (say, 100 nodes), users may only care about robustness with respect to a small subset (e.g., background, texture, lighting). In our latent DAG formalism, if certain spurious variables are never intervened on, then they behave as stable features across domains and can legitimately be exploited by a robust predictor. Thus, we do not require knowledge of all spurious factors—only of those for which robustness is desired.
>
> This is closely analogous to standard supervised learning: when practitioners assign labels $Y$, they do not enumerate all task-relevant features explicitly; rather, labels implicitly tell the model which aspects of $X$ should be predictive. In our case, invariant pairs play the complementary role: they implicitly tell the model which aspects of $X$ should be ignored.
>
> Regarding tabular data, this corresponds to our Level 1 “implicit knowledge” scenario. The right question is: *Robust to what, for this specific application?* For example, in a medical prediction task where robustness to patient ethnicity is desired, one can construct (or approximately match) examples of patients with similar clinical conditions but different ethnicities [11]. These pairs encode exactly the robustness property the practitioner cares about, without requiring an exhaustive catalog of all other possible spurious factors.
>
> To summarize in the context of this comment:
>
> 1. You need a specific robustness target, rather than hoping the model will somehow be robust to arbitrary, unspecified spurious factors.
> 2. Given that target and the application context, you then collect/construct/augment data to form invariant pairs that reflect the particular nuisance factors you want the model to ignore.
>
> Our method is explicitly designed for this targeted, application-driven notion of robustness, not for universal invariance to every possible spurious source.

---

> ### Author Response · Authors · 2025-11-21
> **Rebuttal (Part 2 of 3)**
>
> > The logic of the proposal seems problematic to me...
>
> We appreciate the insightful question and the chance to clarify the connection. We hypothesize that invariant pairs provide a data-centric approach for OOD robustness. Indeed, invariant pairs implicitly provide information about the optimally robust classifier. However, our current theory focuses on a specific type of invariant pairs, i.e., spurious counterfactuals, and proves that they are indeed helpful for OOD robustness. In summary, we start at the higher-level concept of invariant pairs as an abstraction, and then focus on proving things about the spurious counterfactual case of this.
>
> To delve more deeply, as you insightfully noted, all spurious counterfactuals are invariant pairs but not all invariant pairs are spurious counterfactuals.  For example, there could be directions that are irrelevant to the prediction and thus pairs of points along this direction would be invariant but would not help remove spurious directions. Concretely, the linear classifier space can be partitioned into three subspaces:
>
> 1.  The optimally robust subspace (the “good”) - The space of the optimally robust classifier.
> 2.  The spurious subspace (the “bad”) - If the classifier is not orthogonal to this subspace, an intervention on the spurious variables could make the OOD test error arbitrarily large.
> 3. The leftover subspace (the “neutral” subspace) - If a classifier is not orthogonal to this space, the error may be worse than the optimally robust classifier but this space is assumed to not be intervened and thus is stable across train and test.
>
> Invariant pair differences lie within either the “bad” or “neutral” subspace. Because identifying the “neutral” subspace is not required, we focus on identifying the spurious “bad” subspace via counterfactual pairs, whose differences always lie in the spurious subspace. However, invariant pairs that intersect with the neutral subspace should neither increase nor decrease the OOD robustness. Essentially, if such pairs are used, they would implicitly remove part of the neutral subspace, but the optimal classifier (even using ERM) would lie in the remaining subspace. The one caveat in practice is that if the invariant pairs include these “neutral” invariant pairs, the hyperparameter $r$ might have to be set larger to accommodate for the neutral subspace being captured. Ultimately, we do not expect this fundamentally changes any of our empirical or theoretical results but will make sure to clarify this distinction in the final version.
>
> > What is the output space of the robust classifier...
>
> The output of the optimally robust classifier is the label distribution, i.e., probabilities or log probabilities of the classes​​. Thanks for asking this clarification question. We will make it clearer in the final version.
>
> > The examples about the collection of invariant pairs...
>
> Thanks for the insightful question! Because the robust classifier in our definition outputs the label distribution (see above), then knowing the label is *not sufficient* to determine an invariant pair. Additionally, knowing the label is *not necessary*. An example is two x-rays of the same person. Assuming the machines are reasonably equivalent in quality, the uncertainty about the person having a disease is invariant regardless of x-ray machine and regardless of the actual label. Thus, knowing the label is neither necessary nor sufficient to determine invariant pairs. In practice, we could use the label as a weak proxy for the robust prediction probabilities and match samples that have the same class label but different domain labels. Empirically, we have found that this very rough approximation approach is not very effective because the pairs have a high amount of noise (see Table 1,3,4 in the paper on using random pairing with same $Y$ labels).
>
> > Definition 1 is stated for classifiers...
>
> Yes. Theorem 1 already covers standard cross-entropy losses for classification. In particular, the “logistic regression” loss in the theorem is exactly the binary cross-entropy loss used for binary classification. The extension to multiclass softmax cross-entropy is straightforward (e.g., via vector-valued logits) and does not change the structure of the result.
>
> You are correct that Definition 1 is currently phrased only for classifiers. In the revision, we will restate it for general predictors (including both classification and regression), so that its connection to Theorem 1 is explicit and the applicability to cross-entropy classification is clearer. Thanks for pointing this out.

---

> ### Author Response · Authors · 2025-11-21
> **Rebuttal (Part 3 of 3)**
>
> > The authors also claim that one feasible way ...
>
> Thanks for asking. Please note the distinction between the data augmentation itself as a way to *create* examples and methods that *use* these augmented examples during training. Our paper is that data augmentation is one **mechanism to obtain approximate invariant pairs**, not that NCM is equivalent to standard data augmentation.
>
> Concretely, one can use augmentations in at least two ways:
> 1. **Plain augmented ERM:** augment samples and train ERM on the union. This is the most common use of augmentation.
> 2. **Augmentation-based alignment and contrastive learning:** use augmented pairs inside a contrastive or pairwise distance objective. Standard contrastive learning relies heavily on negative pairs to prevent representation collapse. If one removes negative pairs, the objective typically reduces to a pairwise matching term (similar to MatchDG with an $\ell_2$ penalty).
>
> Note that NCM is different from both. We use augmentations (among other mechanisms) to obtain pairs $(x, x')$ that should be invariant, but then we explicitly exploit the *pair structure* to:
> - form the difference matrix of pairs;
> - estimate the spurious subspace via truncated SVD;
> - enforce an **orthogonality constraint** on the classifier with respect to this subspace.
>
> In other words, augmentations provide *inputs*, while NCM specifies a distinct **optimization principle** on top of those inputs. Furthermore, no matter how they are used, data augmentations will only provide robustness to the spurious correlations implied by the augmentations (e.g., rotation augmentations will not make the model robust to semantic spurious correlations like camels are usually on desert backgrounds). Our framework is more targeted to semantic spurious correlations (like in ColoredMNIST or Waterbirds) that cannot be fixed by standard data augmentations. Rather, we use a small number of noisy/approximate counterfactuals to gain robustness.
>
> > ... we have the counterfactual data pairs, why don’t we directly align them using some distance metrics...
>
> Thanks. Our goal is not merely to make paired examples close in representation space, but to ensure that the predictor is orthogonal to the spurious subspace estimated from the pairs. Once the spurious subspace is identified, this orthogonality constraint can be enforced in several ways (e.g., reparameterization, hard projection, Lagrangian penalty, etc.). We chose the projection formulation because, in the linear setting, it both (i) removes the spuriously completely without the need of tuning the Lagrange parameter when data pairs are clean and (ii) is empirically effective. In contrast, pairwise distance or contrastive losses introduce multiple tuning parameters (e.g., temperature, margins, weighting of the pairwise term) and do not directly enforce that the classifier lies in the orthogonal complement of the spurious subspace. They encourage pairs to be close, but they do not guarantee that all directions spanned by the noisy difference matrix are removed. Furthermore, they may unduly penalize non-spurious directions, increasing error. This makes it harder to obtain sharp, finite-sample guarantees of the kind we derive, and it can be less sample-efficient in the noisy few-pair regime we care about.
>
> For nonlinear models, the denoising is no longer as direct or explicit, and we do not claim our current theory covers that regime, and projection is merely work in linear setting only.  However, our linear analysis suggests that a spectrum-aware, sparsity-inducing treatment of the noisy pair differences (akin to an $\ell_1$-type thresholding in the singular values) is more appropriate than a simple $\ell_2$ matching loss. This is conceptually different from MatchDG-style $\ell_2$ penalties: our method explicitly exploits the low-rank spurious structure and its spectral gap. Further, the low-rank spurious should be contributed by the difference of the invariant pairs, not the predictivity of the feature. Otherwise, we are still using spurious features. This also disproves the idea of just using a smaller dimension of latent representation.
>
> > I understand that theoretical analyses...
>
> A: We would like to emphasize that the focus of this paper is the methodology, specifically the proposed approach, algorithm, and theoretical analysis. The primary goal of this work is to introduce the NCM methodology and provide a rigorous, controlled validation of its core principles on established benchmarks. Given that there is currently a lack of large-scale benchmarks with available counterfactual pairs in the literature, we choose PACS and Waterbirds, which are well established datasets, and estimate their counterfactuals in order to compare the domain generalization performance of NCM with existing methods. However, estimating counterfactual pairs for general datasets is an important but separate line of research~[12,13,14] and lies outside the scope of this paper.

---

> > ### Author Response · Authors · 2025-11-21
> > **References**
> >
> > [1] Rosenfeld, Elan, Pradeep Kumar Ravikumar, and Andrej Risteski. "The Risks of Invariant Risk Minimization." International Conference on Learning Representations.
> >
> > [2] Chen, Tianyu, et al. "Identifying general mechanism shifts in linear causal representations." Advances in Neural Information Processing Systems 37 (2024): 42405-42429.
> >
> > [3] Dörfler, Julian, et al. "On the complexity of identification in linear structural causal models." Advances in Neural Information Processing Systems 37 (2024): 100108-100130.
> >
> > [4]Dong, Xinshuai, et al. "On the parameter identifiability of partially observed linear causal models." Advances in Neural Information Processing Systems 37 (2024): 30740-30771.
> >
> > [5] Sanjaroonpouri, Vahideh, and Pouria Ramazi. "Linear SCM Identification in the Presence of Confounders and Gaussian Noise." The Thirteenth International Conference on Learning Representations (Spotlight).
> >
> > [6] Kostin, Julia, Nicola Gnecco, and Fanny Yang. "Achievable distributional robustness when the robust risk is only partially identified." Advances in Neural Information Processing Systems 37 (2024): 83915-83950.
> >
> > [7] Ehyaei, Ahmad-Reza, Golnoosh Farnadi, and Samira Samadi. "Wasserstein distributionally robust optimization through the lens of structural causal models and individual fairness." Advances in Neural Information Processing Systems 37 (2024): 42430-42467.
> >
> > [8] Chen, Jinghui, Yuan Cao, and Quanquan Gu. "Benign overfitting in adversarially robust linear classification." Uncertainty in Artificial Intelligence. PMLR, 2023.
> >
> > [9] Dohmatob, Elvis, and Meyer Scetbon. "Precise accuracy/robustness tradeoffs in regression: Case of general norms." Forty-first International Conference on Machine Learning. 2024.
> >
> > [10] Salaudeen, O. E., Chiou, N., & Koyejo, S. (2024). On domain generalization datasets as proxy benchmarks for causal representation learning. In NeurIPS 2024 Causal Representation Learning Workshop.
> >
> > [11] Horesh, Yair, et al. "Paired-consistency: An example-based model-agnostic approach to fairness regularization in machine learning." Joint European Conference on Machine Learning and Knowledge Discovery in Databases. Cham: Springer International Publishing, 2019.
> >
> > [12] Alaluf, Y., Garibi, D., Patashnik, O., Averbuch-Elor, H., & Cohen-Or, D. (2024, July). Cross-image attention for zero-shot appearance transfer. In ACM SIGGRAPH 2024 conference papers (pp. 1-12).
> >
> > [13] Kolkin, N., Salavon, J., & Shakhnarovich, G. (2019). Style transfer by relaxed optimal transport and self-similarity. In Proceedings of the IEEE/CVF conference on computer vision and pattern recognition (pp. 10051-10060).
> >
> > [14] Hu, Y., Zhuang, C., & Gao, P. (2024, December). DiffuseST: Unleashing the Capability of the Diffusion Model for Style Transfer. In Proceedings of the 6th ACM International Conference on Multimedia in Asia (pp. 1-1).

---

> > > ### Comment · Reviewer_FWNp · 2025-11-25
> > >
> > > Thank you for the detailed responses. Unfortunately, after reviewing them, I find myself more confused. Several points seem internally inconsistent or raise further concerns about the problem formulation, the role of invariant pairs, and the practical applicability of the proposed method.
> > >
> > >  1/ The authors state that the paper targets settings where a specific robustness target is known, and that knowing what we want robustness against should guide the collection of invariant pairs. However, my understanding of the domain generalization (DG) problem is that the aim is to generalize to unseen and unknown target domains. If we already know the target domains and can collect even a small sample from them, the problem becomes much simpler, and existing DG methods can be readily adapted. From my understanding, the reason current DG methods appear to struggle with “arbitrary unspecified spurious factors” is because they attempt to operate under the challenging setting of not knowing the target domain, i.e., lacking the “what” of robustness. In such settings, the best we can hope for is to learn causal invariant features that determine the semantic content of the label.
> > >
> > > &nbsp;
> > >
> > > 2/ Regarding my question about why labels cannot be directly used to collect invariant pairs: your response clarifies that invariant pairs correspond to samples sharing the same label probability distribution under the optimal classifier. Since this distribution is not known in practice, proxies such as class labels or augmentations must be used. Even if labels are not theoretically sufficient or necessary, in realistic scenarios we must rely on them anyways to obtain meaningful invariant pairs.
> > >
> > > This raises a practical concern. In the discussion of DomainBed, you claim that *"there is currently a lack of large-scale benchmarks with available counterfactual pairs”* because collecting or estimating counterfactuals is difficult. It is because it is difficult that we resort to collecting invariant pairs instead. Why not use the above strategies in this case (labels, augmentation or robustness target knowledge)? If benchmarks already struggle to provide suitable pairs, how can we expect real-world applications to be more feasible?
> > >
> > > &nbsp;
> > >
> > > 3/ The authors acknowledge that invariant pairs are not necessarily counterfactual pairs and therefore do not always serve the same purpose. However, the responses do not clearly address the fundamental issue: why invariant pairs should be expected to substitute for counterfactual pairs in enabling generalization. From what I gathered, invariant pairs are useful only under specific—and potentially atypical—conditions. This substantially limits the impact of the proposed contribution. I initially had high expectations for the method because if one could reliably obtain invariant pairs that imply counterfactual pairs, the approach would offer a tractable surrogate for a notoriously hard problem, which would indeed constitute a significant contribution. Without this, I am less convinced by the contribution. Additionally, the technical part of the optimization objective seems incremental relative to existing methods.
> > >
> > > &nbsp;
> > >
> > > 4/ Given the points above, I remain unclear on why the proposed method is preferable to existing approaches. The justification provided against contrastive methods only works in linear settings, and as the authors claim, the theoretical guarantees offered in the paper may not hold in more complex situations. This leaves open the question of when—and whether—the method offers a clear advantage in practice.
> > >
> > > ---
> > > Due to the reasons above, I retain my rating. However, I strongly encourage the authors to reconsider my points to improve the work, as I really like the motivation of the paper, which I believe would give a great contribution if the problem can be addressed thoroughly.

---

> ### Author Response · Authors · 2025-11-27
> **Reply (Part 1|3)**
>
> Thanks for the follow-up so we can clarify. There is a misunderstanding of our work.
> >  The authors state that the paper targets settings where a specific robustness target is known, and that knowing what we want robustness against should guide the collection of invariant pairs.
>
> We perfectly align with the standard DG setup you are referring to. We do not require any sample from the target domain nor we need to know the specific target domain distribution. As formalized in Definition 6, we define a class of domains $\mathcal{E}$ that captures the types of distribution shifts we care about. The training domains form a subset of this class, and as stated in Definition 1, the goal is to learn a classifier that performs well on the worst-case domain $e \in \mathcal{E}$. This worst-case domain is not one of the training domains giving finite training domains we see, and all we use is samples from the training domain (In page 5,line 216, we explicitly state “Noisy Counterfactual Matching (NCM). Given a set of CF pairs solely from the **training domains**), thus we do not use any information from the test domains. Instead, we explicitly aim to be provably robust to the worst-case domain at test time. **Implicitly** knowing which features are spurious is precisely what allows us to properly specify the class $\mathcal{E}$ of domains we care about. This is fundamental for formulating domain generalization: the distribution shifts of interest in the target domains must be reflected in the differences among the training domains (see Definition 6). Any multi-domain DG task necessarily makes an assumption that the training domain labels *implicitly* encode spurious features. In existing benchmarks, this assumption is often not stated explicitly, but it is built into the domain setup.
> For example, in the RotatedMNIST benchmark, the training domains contain digits under several rotations, and the test domains ask us to generalize to an unseen rotation. This is a well-defined DG problem, because the test-time shift (rotation) is aligned with the distribution shifts across training domains. In contrast, if the test domain consisted of digits with different colors, or even images of animals instead of digits, this would fall outside the shifts encoded by the training domains and would correspond to an ill-defined DG problem. That is to say: the training domains implicitly encode spurious features (in this case, the rotation, not the color nor other features). But we do not need to know the specific rotation level of the test domain.
>
> >  In such settings, the best we can hope for is to learn causal invariant features that determine the semantic content of the label.
>
> We agree that, in principle, the ideal goal in the aforementioned DG setting is to learn causal invariant features that determine the semantic content of the label. However, reliably learning such features from finite multi-domain labeled data itself requires Assumptions which are either hard to satisfy or impossible to verify. For example, existing theoretical guarantees for recovering causal invariants typically rely on counterfactual inference with interventional data [1,2,3,4]. These works usually have more assumptions, like atomic interventions, positivity, faithfulness etc. These strong assumptions are rarely satisfied in standard DG benchmarks or realistic applications. Thus, this “best we can hope for” objective, probably identifying causal invariant features, is very challenging under practical assumptions. Our contribution is to show that certain forms of counterfactual supervision, i.e., noisy invariant pairs constructed from training domains, offer a promising and more realistic alternative. Rather than fully identifying the causal features, we use a small number of approximate counterfactual pairs to regularize the robust classifier, and we provide theoretical guarantees for robustness within the specified class of shifts.
>
> [1] Schölkopf, Bernhard, et al. "Toward causal representation learning." Proceedings of the IEEE 109.5 (2021): 612-634.
>
> [2] Deng, Bin, and Kui Jia. "Counterfactual supervision-based information bottleneck for out-of-distribution generalization." Entropy 25.2 (2023): 193.
>
> [3] Squires, Chandler, et al. "Linear causal disentanglement via interventions." International conference on machine learning. PMLR, 2023.
>
> [4] Zhang, Jiaqi, et al. "Identifiability guarantees for causal disentanglement from soft interventions." Advances in Neural Information Processing Systems 36 (2023): 50254-50292.

---

> ### Author Response · Authors · 2025-11-27
> **Reply (Part 2|3)**
>
> > The authors acknowledge that invariant pairs are not necessarily counterfactual pairs and therefore do not always serve the same purpose...
>
> Thanks for raising this point.  We do not claim that these conditions hold in every DG problem. Instead, we argue that they capture a broad and practically important regime where the practitioner could know what type of variability should be treated as nuisance and can obtain approximate “same-object, different-nuisance” pairs. Typical examples include: repeated measurements of the same patient under different acquisition settings, multiple views of the same scene under different lighting or style, or synthetic style/background changes applied to the same underlying object. In these settings, labels alone do not tell us which factor changed between two samples, but labels plus pairing structure do, and our theory explains how such pairs can be used to recover the same invariances as ideal counterfactual supervision.
>
> > From what I gathered, invariant pairs are useful only under specific...
>
> We first clarify that invariant pairs **between domains** (i.e., $(x\_e, x’\_{e’})$, where $x$ and $x’$ have domain labels $e \neq e’$) are the types of invariant pairs that are approximate spurious counterfactuals and provide useful signals for OOD robustness. As explained above, a change in domain label *implicitly* defines a spurious change in the environment—the same type of assumption that standard DG methods implicitly make. And, we use these cross-domain invariant pairs in our experiments. Thus, if domain labels are known (as assumed in DG), then we aim to collect invariant pairs *within the training domains* that have different domain labels. Thus, we respectfully disagree that these are “specific—and potentially atypical” conditions but rather based on common conditions (implicitly) assumed by DG. Further, we explained multiple feasible ways to obtain such useful invariant pairs in practice (See Appendix A.2 p.15).
>
> Secondly, we specifically seek to make this new paradigm practical by considering two key practical challenges: (1) noise in estimating these cross-domain invariant pairs that approximate spurious counterfactuals and (2) a small number of pairs. In fact, these are our main contributions that provide a first theoretically-grounded yet practical alternative to the standard DG setup for OOD robustness. Our results show that, in this regime, invariant pairs do provide a surrogate for counterfactual supervision with favorable sample requirements (e.g., $K$ pairs versus $K$ domains for IRM-type methods), which is proven empirically in our experiments. Given all this, we actually agree that we offer a “tractable surrogate for a notoriously hard problem, which would indeed constitute a significant contribution.”  Indeed, we believe it opens up a new frontier, with both its promises and challenges.
>
>
>
> >Additionally, the technical part of the optimization objective seems incremental relative to existing methods.
>
> Finally, on the technical side, we agree that the optimization objective is deliberately simple and resembles existing regularization-based methods. The novelty is not in introducing a complex new architecture or loss, but in the way the supervision is structured (through counterfactual-style pairs) and in the accompanying theory that justifies this structure as a principled surrogate for counterfactual robustness. Simplicity is our strength: our method can be implemented by minimal modifications to standard training pipelines, yet is supported by new theoretical guarantees about when and why invariant pairs enable OOD generalization. Our paper provides the first non-asymptotic bound of how counterfactual pairs sample size and noises could affect DG error quantitatively.

---

> ### Author Response · Authors · 2025-11-27
> **Reply (Part 3|3)**
>
> > ... I remain unclear on why the proposed method is preferable ...
>
> Our theory provides provable robustness guarantees in the linear setting we analyze, and empirically we show that our method achieves significantly stronger OOD robustness than existing approaches on the challenging ColoredMNIST benchmark. While our formal guarantees currently focus on linear models, prior nonlinear experiments such as MatchDG and DIRT already demonstrate that pairing-based approaches can achieve strong empirical performance. Our work is complementary to these methods: it provides a formal framework that explains **why** this type of supervision can improve robustness even with few-shot, noisy pairs, and clarifies the conditions under which pairing-based supervision serves as a surrogate for counterfactual information.
>
> We respectfully ask the reviewer to reconsider our work in this light: as both a technically grounded contribution on provable robustness and a step toward a frontier, data-centric approach to OOD generalization. We fully acknowledge that many open questions remain in this direction. However, we view the use of estimated invariant pairs as a **strength** rather than a limitation: it broadens the design space beyond the standard paradigm in which robustness is pursued only through labeled multi-domain samples. Our results suggest that pairing-based supervision is a viable and principled alternative, and we point out multiple scenarios where collecting such pairs is feasible. We hope this paper will help stimulate further research on data collection and representation design for OOD robustness. Instead of viewing the need to collect pairs as a drawback, we see it as an opportunity to explore a complementary data-collection strategy: why should OOD robustness be pursued only through multi-domain labeled samples, and not also through carefully constructed invariant pairs?

---

### Official Review · Reviewer_vnFv · 2025-10-24

**Soundness:** 2
**Presentation:** 3
**Contribution:** 2
**Rating:** 4
**Confidence:** 2

**Summary:**

The paper presents Noisy Counterfactual Matching (NCM), a method for domain generalization. NCM uses pairs of samples that differ only in spurious aspects (features) to enforce consistent predictions and filters out spurious directions with truncated SVD. The authors provide theoretical support showing that few diverse pairs are sufficient for invariance and report experiments on ColoredMNIST, Waterbirds-CF, and PACS where NCM matches or slightly outperforms established baselines such as IRM, GroupDRO, SWAD, and LISA.

**Strengths:**

The paper presents a an approach to domain generalization based on counterfactual matching. Its originality lies in reformulating invariant representation learning without relying on explicit domain labels, supported by a well-developed theoretical framework. The analysis is technically sound, with proofs that connect counterfactual consistency to invariance and empirical results that align with the theory. The experimental setup is appropriate, using standard benchmarks.

**Weaknesses:**

While the paper is well-executed, its novelty is somewhat limited relative to prior work on counterfactual and invariant learning, such as MatchDG (ICLR 2021) . The distinction between NCM and these methods, beyond the use of formal proofs, could be explained more clearly. The empirical evaluation, though thorough on benchmark datasets, depends on synthetic or curated counterfactuals (e.g., Waterbirds-CF), which limits evidence of real-world applicability. Demonstrating how counterfactuals could be obtained or approximated in realistic settings would strengthen the paper. The study would also benefit from testing on additional or more recent benchmarks.

**Questions:**

1. The experiments primarily rely on synthetic or semi-synthetic datasets where counterfactuals are artificially constructed. How would NCM perform on real-world datasets where counterfactual pairs are unavailable or difficult to define or noisy?
2. Would the authors consider including results for baselines on the synthetic dataset to confirm that NCM’s advantages are not specific to its own simulation setup?
3. Please include 2023, 2024, and 2025 methods in Related Work and, where feasible, add a conceptual and empirical comparison.

---

> ### Author Response · Authors · 2025-11-16
> **Clarification question**
>
> > Please include 2023, 2024, and 2025 methods in Related Work and, where feasible, add a conceptual and empirical comparison.
>
> Which specific research works are you referring to? We are happy to include and discuss related works that we have missed.

---

> > ### Comment · Reviewer_vnFv · 2025-11-17
> >
> > I wanted to note that majority of the baselines included in the experiments are from 2021 or earlier. I also acknowledge that I initially overlooked that LISA is from 2024, which is already a strong and recent baseline. I would prefer not to suggest specific baselines myself, as it is the authors’ responsibility to identify and position their work within the most up-to-date state of the art.

---

> ### Author Response · Authors · 2025-11-21
> **Rebuttal (Part 1 of 2)**
>
> We thank you for the reviewer's feedback and would like to address the concern below.
>
> > While the paper is well-executed, its novelty ...
>
> > The experiments primarily rely on synthetic ...
>
> We respectfully disagree that our contribution is only marginally novel relative to prior counterfactual/invariant learning work such as MatchDG, or that our evaluation provides limited evidence of applicability. Below, we address the reviewer’s four concerns.
>
> ## Conceptual distinction from MatchDG: noisy, few-shot counterfactual pairs.
> MatchDG assumes access to effectively clean, well-aligned cross-domain pairings and does not provide a finite-sample analysis under noisy pairs. In contrast, our work is explicitly designed for the noisy, few-shot counterfactual regime and provides a generalization bound that characterizes the error as a function of both the noise level and the number of pairs (see Summary of Contributions). Relaxing the oracle-pair assumption and deriving finite-sample guarantees for noisy counterfactuals is not a minor variant but an important step that brings this line of ideas from idealized settings closer to practical applicability with theoretical guarantees. Conceptually, the denoising using truncated SVD versus $\ell_2$ penalty on the difference between pair difference  in our setting is analogous to the step from ridge to lasso (moving from an $\ell_2$ to an $\ell_1$ constraint): a seemingly modest change in formulation that enables a qualitatively different behavior and has historically led to substantial developments in the literature.
>
> ## Regarding linearity
> Given our theoretical analysis and the utility of linear probing on top of representation models in practice, we argue that our linearity assumption is not unusually restrictive. Rather, it helps us isolate core concepts and provide the foundation for non-linear analysis, which we agree is an important future research direction. We explain more below:
>
> 1. **Linear SCMs are a canonical theoretic testbed isolating the core challenges.**
> Linear SCMs [1-9] are a standard starting point in causal and domain-generalization theory: they allow one to isolate some other challenges to obtain sharp, interpretable results about identifiability and robustness. In our setting, the linear SCM assumption lets us precisely characterize how the spurious subspace relates to counterfactual differences and when a small set of invariant pairs suffices to recover the robust predictor. For example, ColoredMNIST might be thought of as an overly easy task in the era of LLM, it is indeed shown to be a very hard generalization task. [10]
>
> 2. **The results are about geometry, not just about “toy” models.**
> Our guarantees are fundamentally geometric: they relate out-of-domain error to (i) the alignment between the subspace spanned by invariant-pair differences and the true intervention subspace, and (ii) spectral properties of the corresponding covariance. This provides conceptual guidance that extends beyond strictly linear generative processes. For example, the same geometric picture underlies our use of NCM as a linear probe on top of powerful nonlinear encoders (e.g., CLIP) in the experiments.
> 3. **Theory plus practice: non-linear feature maps in experiments.**
> While the SCM analysis is linear, our empirical evaluation is explicitly non-linear: we apply NCM to CLIP features and other learned representations. In this sense, the theory clarifies why the algorithm should be effective once an approximately linear “task+spurious” structure is exposed in feature space, and the experiments demonstrate that this intuition carries over to realistic, high-dimensional settings.
> 4. **Scope and future extensions.**
> We fully acknowledge that extending the analysis to more general (e.g., nonlinear or non-additive) SCMs is an important direction for future work. Our goal in this paper is to establish a first, clean characterization of how invariant pairs control robustness under interventions, and to show that even in the linear case this already yields nontrivial, practically useful insights and algorithms.

---

> ### Author Response · Authors · 2025-11-21
> **Rebuttal (Part 2 of 2)**
>
> (Continue on the first question)
>
> **On additional or more recent benchmarks.**
> We do not add more benchmark datasets mainly because most DG benchmarks *do not provide any structure for constructing counterfactual pairs*, making them unsuitable for evaluating a pair-based method such as NCM. Instead, we focus on representative datasets where counterfactuals can be empirically estimated. In particular, we choose ColoredMNIST to stress-test our theory. ColoredMNIST is widely regarded as harder than many DG benchmarks due to the strong color–label spurious correlation and the “inverse-line” evaluation [10]. We emphasize that domain count, dataset size, or resolution should not be used as proxies for problem difficulty. Even with only three domains, ColoredMNIST can present a harder scenario when the two training domains are similar and the test domain differs markedly, than a setting with many training/test domains that are mutually similar.
>
> By contrast, many high-dimensional DG benchmarks (e.g., DomainBed [11], WILDS [12]) have been shown to exhibit the “accuracy-on-the-line’’ phenomenon [13], where most DG methods fail to outperform ERM. As argued in [10], ColoredMNIST instead shows an “accuracy-on-the-reverse-line’’ effect and is therefore better suited for evaluating robustness to spurious correlations in a near worst-case scenario. For this reason, we deliberately prioritize benchmarks that reflect the specific robustness challenge NCM is designed to address, rather than simply adding more datasets where counterfactual structure is absent.
>
> **On availability and definition of counterfactual pairs.**
> We agree that the availability and quality of counterfactual (or invariant) pairs is crucial. This is discussed in Appendix A.2, where we describe several practical scenarios. In order to meaningfully target robustness beyond the training distribution, one must have at least some information about which variations should be treated as invariant. This information can then be used to define approximate pairs, even implicitly. For example, in medical imaging, explicit pixel-level counterfactuals may be hard to define, but it is often feasible to collect or construct pairs that differ in specific acquisition conditions or visible artifacts (e.g., medical devices) while preserving the underlying diagnosis. These settings naturally yield approximate noisy invariant pairs of the kind NCM is designed to leverage. We will move a concise summary of this discussion from the appendix into the main text to make these practical scenarios more visible.
>
> > Would the authors consider including results for baselines ...
>
> Please see the table below for the synthetic results for other method. We carefully include all the implementation details and code for our synthetic dataset. If any part might seems to be “favor” our method, please point it out and we’d like to address your concern.
>
> | $\epsilon=5$ | In-domain | DG   |
> |--------------|-----------|------|
> | ERM          | 0.99      | 0.31 |
> | IRM          | 0.98      | 0.29 |
> | REx          | 0.98      | 0.31 |
> | GroupDRO     | 0.99      | 0.35 |
> | Fish         | 0.99      | 0.37 |
> | SWAD         | 0.97      | 0.30 |
> | LISA         | 0.98      | 0.33 |
> | MatchDG      | 0.89      | 0.49 |
> | NCM          | 0.86      | 0.65 |
>
> The spurious feature here is predictive thus most of the methods perform similar to ERM, which is not robust. MatchDG works better but is not as effective at ignoring the noise in the counterfactuals compared to NCM.
>
> > Please include 2023, 2024, and 2025 ...
>
> First, we note that our goal in the Related Work section is to cover *representative and the conceptually closest* methods, rather than to provide a comprehensive list of papers. In the current version, we already discuss key lines of work, including MatchDG, DIRT, invariant representation learning methods, and data-augmentation–based approaches with LISA, published in 2024. From our understanding, we did not miss any major lines of related work (though we are happy to include any specific papers we have missed).
>
> Nonetheless, according to the referee’s suggestion, we have aimed to find more recent works to include. Specifically, we will add references to the following recent DG works using data augmentation [14-17].

---

> > ### Author Response · Authors · 2025-11-21
> > **References**
> >
> > [1] Rosenfeld, Elan, Pradeep Kumar Ravikumar, and Andrej Risteski. "The Risks of Invariant Risk Minimization." International Conference on Learning Representations.
> >
> > [2] Chen, Tianyu, et al. "Identifying general mechanism shifts in linear causal representations." Advances in Neural Information Processing Systems 37 (2024): 42405-42429.
> >
> > [3] Dörfler, Julian, et al. "On the complexity of identification in linear structural causal models." Advances in Neural Information Processing Systems 37 (2024): 100108-100130.
> >
> > [4]Dong, Xinshuai, et al. "On the parameter identifiability of partially observed linear causal models." Advances in Neural Information Processing Systems 37 (2024): 30740-30771.
> >
> > [5] Sanjaroonpouri, Vahideh, and Pouria Ramazi. "Linear SCM Identification in the Presence of Confounders and Gaussian Noise." The Thirteenth International Conference on Learning Representations (Spotlight).
> >
> > [6] Kostin, Julia, Nicola Gnecco, and Fanny Yang. "Achievable distributional robustness when the robust risk is only partially identified." Advances in Neural Information Processing Systems 37 (2024): 83915-83950.
> >
> > [7] Ehyaei, Ahmad-Reza, Golnoosh Farnadi, and Samira Samadi. "Wasserstein distributionally robust optimization through the lens of structural causal models and individual fairness." Advances in Neural Information Processing Systems 37 (2024): 42430-42467.
> >
> > [8] Chen, Jinghui, Yuan Cao, and Quanquan Gu. "Benign overfitting in adversarially robust linear classification." Uncertainty in Artificial Intelligence. PMLR, 2023.
> >
> > [9] Dohmatob, Elvis, and Meyer Scetbon. "Precise accuracy/robustness tradeoffs in regression: Case of general norms." Forty-first International Conference on Machine Learning. 2024.
> >
> > [10] Salaudeen, O. E., Chiou, N., & Koyejo, S. (2024). On domain generalization datasets as proxy benchmarks for causal representation learning. In NeurIPS 2024 Causal Representation Learning Workshop.
> >
> > [11] Gulrajani, I., & Lopez-Paz, D. (2020). In search of lost domain generalization. arXiv preprint arXiv:2007.01434.
> >
> > [12] Sagawa, S., Koh, P. W., Lee, T., Gao, I., Xie, S. M., Shen, K., ... & Liang, P. (2021). Extending the wilds benchmark for unsupervised adaptation. arXiv preprint arXiv:2112.05090.
> >
> > [13] Miller, John P., et al. "Accuracy on the line: on the strong correlation between out-of-distribution and in-distribution generalization." International conference on machine learning. PMLR, 2021.
> >
> > [14] Chien, Jen-Tzung, Mahdin Rohmatillah, and Chang-Ting Chu. "Strategic Optimization for Worst-Case Augmentation and Classification." IEEE/ACM Transactions on Audio, Speech, and Language Processing (2024).
> >
> > [15] Miao, Qiaowei, Yawei Luo, and Yi Yang. "DICS: Find Domain-Invariant and Class-Specific Features for Out-of-Distribution Generalization." ICASSP 2025-2025 IEEE International Conference on Acoustics, Speech and Signal Processing (ICASSP). IEEE, 2025.
> >
> > [16] Jiang, Ziyi, et al. "Cbda: Contrastive-based data augmentation for domain generalization." IEEE Transactions on Computational Social Systems (2024).
> >
> > [17] Liu, Yingnan, et al. "Cross-domain feature augmentation for domain generalization." arXiv preprint arXiv:2405.08586 (2024).

---

### Official Review · Reviewer_zQ3E · 2025-10-26

**Soundness:** 2
**Presentation:** 3
**Contribution:** 2
**Rating:** 4
**Confidence:** 4

**Summary:**

This paper proposes a data-centric approach to domain generalization called Noisy Counterfactual Matching (NCM). Instead of enforcing representation-level invariance, the authors suggest learning invariance directly from invariant data pairs. The method approximates counterfactual sample pairs and constrains the classifier so that its predictions remain consistent across them. Theoretical analysis shows that when these counterfactual pairs approximate the true invariant directions, the classifier achieves provable robustness to spurious correlations. Experiments on synthetic data show modest improvements over baselines.

**Strengths:**

1. Addressing spurious correlations and domain generalization remains an important and open challenge.
2. The paper is well-written and conceptually easy to follow, with a consistent flow from motivation to theory to experiments.
3. Experimental results show modest but consistent improvements on benchmark datasets.

**Weaknesses:**

1. The core idea of enforcing invariance through matching or orthogonality constraints is not entirely new. Previous works such as MatchDG, IRMv1 have already explored related ideas. NCM’s main distinction, using noisy counterfactual pairs, is interesting but incremental and mostly limited to linear settings.
2. The theoretical guarantees hold only for linear models with idealized “counterfactual pairs.” In realistic nonlinear cases (e.g., deep networks), the method lacks justification. The authors claim empirical robustness, but this cannot be directly attributed to the theory presented.
3. The approach assumes access to approximate counterfactual pairs or noisy invariant pairs, which may not be available in practice. The paper does not discuss how these pairs could be obtained in realistic settings or how sensitive the method is to poor-quality matches.
4. While the results show slight gains, they are relatively small, raising questions about whether the added complexity is justified compared to simpler ERM or representation-based baselines.
5. The extension of the orthogonality constraint via SVD projection to deep models is mentioned but not clearly demonstrated or validated experimentally.

**Questions:**

See Weaknesses Part.

---

> ### Author Response · Authors · 2025-11-21
> **Rebuttal (Part 1 of 3)**
>
> Thanks the reviewer for their feedback. We address your questions and concerns below.
>
> > The core idea of enforcing invariance ...
>
> Thank you for raising the concern, we respectfully disagree with the reviewer’s assessment that our contribution is incremental. Below, we address the three concerns separately. Note that IRM and MatchDG belong to different methodological families, so we discuss them separately .
>
> ## Difference compared to IRM
>
> We agree that our work is related to methods that enforce invariance, but our formulation is fundamentally different from IRM-type approaches. IRM-style methods (and related distribution-matching approaches) aim to learn an invariant representation by matching distributions such as $p(h(x))$, $p(h(x)\mid y)$, or $p(y\mid h(x))$ across multiple environments, which is a type of distribution invariance. In contrast, we work with a small set of counterfactual pairs $(x, x')$ and directly constrain the predictor so that its outputs are invariant on these pairs—a type of point-wise invariance. Thus, our pair invariance is enforced using different data and different mechanisms compared to invariant *representation* approaches like IRM. Technically, we estimate the spurious subspace from pair differences and enforce an orthogonality constraint on the classifier; this yields a data-centric, pair-based formulation rather than a distribution-matching representation method.
>
> ## Noisy finite counterfactual pairs: beyond MatchDG.
>
> MatchDG’s theory guarantee assumes access to effectively clean, well-aligned cross-domain pairings and does not provide a finite-sample analysis for noisy pairs. Our work explicitly addresses the *noisy, finite* counterfactual regime and provides a generalization bound that characterizes the error as a function of the noise level and the number of pairs (see Summary of Contributions). Relaxing the oracle-pair assumption and providing finite-sample guarantees for noisy counterfactuals is not a minor variant, but an important step that brings this line of ideas from idealized theory closer to practical applicability with guarantees. The denoising step in our setting is conceptually similar to the step from ridge regression to lasso regression, moving from an $\ell_2$ to an $\ell_1$ constraint: such seemingly modest changes in formulation have historically led to substantial and influential developments in the literature.
>
> ## Regarding limits of the linearity
>
> Given our theoretical analysis and the utility of linear probing on top of representation models in practice, we argue that our linearity assumption is not unusually restrictive. Rather, it helps us isolate core concepts and provide the foundation for non-linear analysis, which we agree is an important future research direction. We explain more below:
> 1. **Linear SCMs are a canonical theoretic testbed isolating the core challenges.**
> Linear SCMs [1-9] are a standard starting point in causal and domain-generalization theory: they allow one to isolate some other challenges to obtain sharp, interpretable results about identifiability and robustness. In our setting, the linear SCM assumption lets us precisely characterize how the spurious subspace relates to counterfactual differences and when a small set of invariant pairs suffices to recover the robust predictor. For example, ColoredMNIST might be thought of as an overly easy task in the era of LLM, it is indeed shown to be a very hard generalization task. [10]
> 2. **The results are about geometry, not just about “toy” models.**
> Our guarantees are fundamentally geometric: they relate out-of-domain error to (i) the alignment between the subspace spanned by invariant-pair differences and the true intervention subspace, and (ii) spectral properties of the corresponding covariance. This provides conceptual guidance that extends beyond strictly linear generative processes. For example, the same geometric picture underlies our use of NCM as a linear probe on top of powerful nonlinear encoders (e.g., CLIP) in the experiments.
> 3. **Theory plus practice: non-linear feature maps in experiments.**
> While the SCM analysis is linear, our empirical evaluation is explicitly non-linear: we apply NCM to CLIP features and other learned representations. In this sense, the theory clarifies why the algorithm should be effective once an approximately linear “task+spurious” structure is exposed in feature space, and the experiments demonstrate that this intuition carries over to realistic, high-dimensional settings.
> 4. **Scope and future extensions.**
> We fully acknowledge that extending the analysis to more general (e.g., nonlinear or non-additive) SCMs is an important direction for future work. Our goal in this paper is to establish a first, clean characterization of how invariant pairs control robustness under interventions, and to show that even in the linear case this already yields nontrivial, practically useful insights and algorithms.

---

> ### Author Response · Authors · 2025-11-21
> **Rebuttal (Part 2 of 3)**
>
> > The theoretical guarantees hold only ...
>
> We respectfully disagree with the reviewer’s characterization of our theory and its relation to the experiments. Since the concern about linearity is addressed in the last question, here we focus on other questions.
>
> ## Idealized counterfactual pairs
> The comment about “idealized counterfactual pairs” does not accurately describe our setting. A central contribution of our paper is to analyze noisy, few-shot counterfactual pairs and to provide generalization guarantees that depend explicitly on the number of pairs and their noise level (see Summary of Contributions). Our generalization bounds make precise how imperfect estimation of the spurious subspace caused by noisy and limited counterfactual data affects out-of-domain error.
>
> ## Our experiments reflect the theory presented
> In the experiments other than the synthetic experiment, the learned part of the model matches our theoretical setting exactly: we freeze the pretrained CLIP encoder and train a linear classifier on top of the frozen features using NCM (i.e., linear probing fine tuning using NCM). In these cases, our analysis applies directly to the classifier head, and the observed robustness improvements are consistent with the mechanism studied in the theory.
>
> > The approach assumes access to approximate counterfactual pairs ...
>
> We kindly disagree with the claim that the paper does not address the practicality of obtaining approximate counterfactual pairs or the sensitivity to their quality.
>
> ## Practical availability of approximate counterfactual / invariant pairs.
> Contrary to the reviewer’s statement, we explicitly discuss when and how such pairs can be obtained. In the main text and Appendix A.2 (“Availability of invariant pairs”), we provide a taxonomy (Table 2) of realistic scenarios, from explicit knowledge of spurious features to purely implicit knowledge, and outline concrete data-collection mechanisms in each case. Examples include:
> 1. medical imaging, where domain experts know that certain devices or artifacts (e.g., fluid lines) should not affect the diagnosis, so paired x-rays with and without these artifacts or simple editing / augmentation can be used to construct approximate counterfactuals;
> 2. multi-device measurement setups (e.g., two microscopes), where repeated measurements of the same sample naturally yield cross-domain pairs. We will move part of this discussion from the appendix into the main text in the revision to make this clearer.
>
> ## Sensitivity to noisy / poor-quality matches is analyzed explicitly.
>
> Our theory is precisely designed to capture the effect of noisy, finite counterfactual pairs. Section 3 formalizes noisy counterfactual differences via the matrix $\Delta_x$, and the generalization bound in Theorem 1 and Corollary 3 shows that test-domain error decomposes into (i) an in-domain term and (ii) a term depending on the estimation error of the spurious subspace, which in turn depends on the noise level and the number and diversity of pairs (through Wedin’s $\sin\Theta$theorem and the singular values of $\Delta_x$). Thus, the method’s sensitivity to poor-quality matches is not left unspecified; it is made explicit in the bound.
> ## Empirical behavior under noisy, few-shot pairs.
> The synthetic experiments in Section 5 directly study the effect of both noise and the number of counterfactual pairs: we compare oracle vs noisy pairs, vary the number of pairs, and observe that (i) a small number of diverse pairs suffices to improve robustness, and (ii) performance degrades gracefully as noise increases, while still outperforming ERM in regimes where the ideal technical conditions are not strictly met.
>
> ## Data complexity
> IRM requires the number of domains  to scale with the spurious feature dimension [1] (i.e., $E > \mathcal{I}\_{\mathcal{F}} $), which may be infeasible or too costly in practice. Our NCM, on the other hand, only requires the number of counterfactual pairs to scale with the spurious feature dimension (i.e., $K \geq \mathcal{I}\_{\mathcal{F}} $), which may be significantly more feasible.
>
> Overall, in contrast to the reviewer’s statement, our work does not assume unrealistically perfect counterfactual pairs; Further it explicitly models noisy, finite pairs, provides guarantees that depend on their quality, and discusses practically motivated scenarios in which such pairs can be collected.

---

> ### Author Response · Authors · 2025-11-21
> **Rebuttal (Part 3 of 3)**
>
> > While the results show slight gains, they are relatively small ...
>
> We do not agree that the observed gains are “slight” or that they fail to justify the added complexity.
>
> ## ColoredMNIST
> On ColoredMNIST, the improvement over ERM is not marginal. This benchmark is well known to be challenging for domain generalization despite its simple images, precisely because of the strong spurious color–label correlation and the “inverse line” evaluation [10]. In this setting, NCM closes a substantial portion of the domain gap relative to ERM. Calling these gains “slight” does not accurately reflect the scale of improvement on a benchmark explicitly designed to stress-test robustness.
>
> ## Realistic datasets (Waterbirds, PACS).
> On more realistic datasets, the improvements are also non-trivial. We obtain roughly **+3%** over ERM on Waterbirds and about **+1.8%** on PACS. Given that the domain generalization gap (difference between in-domain and out-of-domain accuracy) on these benchmarks is typically in the single-digit range, reducing this gap by 2–3% points represents a meaningful improvement. In the DG literature, gains of this magnitude over strong ERM baselines could not be considered negligible.
>
> ## Complexity vs benefit.
> Finally, the additional complexity introduced by NCM is modest: we keep the same backbone and classifier architecture as ERM and add a lightweight constraint based on the SVD of pair differences. There are no extra networks or large auxiliary modules. In view of the consistent improvements across ColoredMNIST, Waterbirds, and PACS, we believe this level of added complexity is well justified.
>
>
> > The extension of the orthogonality constraint via SVD projection to deep models ...
>
> We believe there is a misunderstanding of what we actually claim regarding extensions to deep models. Our theoretical development and the SVD-based orthogonality constraint are formulated for linear predictors. We do not claim a general theoretical extension of this orthogonality constraint with guarantees to arbitrary deep networks.
> In the empirical experiments, we do so in a way that is fully consistent with the theory: we freeze the deep encoder and apply NCM at the linear head on top of the fixed features. In this setting, the learned component is linear, and the SVD-based projection is exactly the mechanism analyzed in our theory; these experiments provide a direct empirical validation of the linear-probing finetuning case.
>
> [1] Rosenfeld, Elan, Pradeep Kumar Ravikumar, and Andrej Risteski. "The Risks of Invariant Risk Minimization." International Conference on Learning Representations.
>
> [2] Chen, Tianyu, et al. "Identifying general mechanism shifts in linear causal representations." Advances in Neural Information Processing Systems 37 (2024): 42405-42429.
>
> [3] Dörfler, Julian, et al. "On the complexity of identification in linear structural causal models." Advances in Neural Information Processing Systems 37 (2024): 100108-100130.
>
> [4]Dong, Xinshuai, et al. "On the parameter identifiability of partially observed linear causal models." Advances in Neural Information Processing Systems 37 (2024): 30740-30771.
>
> [5] Sanjaroonpouri, Vahideh, and Pouria Ramazi. "Linear SCM Identification in the Presence of Confounders and Gaussian Noise." The Thirteenth International Conference on Learning Representations (Spotlight).
>
> [6] Kostin, Julia, Nicola Gnecco, and Fanny Yang. "Achievable distributional robustness when the robust risk is only partially identified." Advances in Neural Information Processing Systems 37 (2024): 83915-83950.
>
> [7] Ehyaei, Ahmad-Reza, Golnoosh Farnadi, and Samira Samadi. "Wasserstein distributionally robust optimization through the lens of structural causal models and individual fairness." Advances in Neural Information Processing Systems 37 (2024): 42430-42467.
>
> [8] Chen, Jinghui, Yuan Cao, and Quanquan Gu. "Benign overfitting in adversarially robust linear classification." Uncertainty in Artificial Intelligence. PMLR, 2023.
>
> [9] Dohmatob, Elvis, and Meyer Scetbon. "Precise accuracy/robustness tradeoffs in regression: Case of general norms." Forty-first International Conference on Machine Learning. 2024.
>
> [10] Salaudeen, O. E., Chiou, N., & Koyejo, S. (2024). On domain generalization datasets as proxy benchmarks for causal representation learning. In NeurIPS 2024 Causal Representation Learning Workshop.

---

> ### Comment · Area_Chair_7kan · 2025-11-27
> **Please reply to the authors' rebuttal**
>
> Dear Reviewer,
>
> The authors have provided their rebuttal. Please reply to it before the rebuttal period ends. Thanks!
>
> Best regards,
>
> AC

---

### Official Review · Reviewer_PwUU · 2025-11-01

**Soundness:** 3
**Presentation:** 3
**Contribution:** 3
**Rating:** 8
**Confidence:** 3

**Summary:**

The paper proposes a data‑centric approach to robustness under domain shift that leverages invariant data pairs (instead of learning invariant representation)—pairs of inputs that should receive the same prediction under a robust classifier. The authors formalize the  spurious counterfactuals in a causal perspective, and prove that spurious counterfactuals are invariant pairs. They introduce Noisy Counterfactual Matching (NCM): augment ERM with a linear constraint that projects predictions onto the orthogonal complement of the spurious subspace estimated via the truncated SVD of pairwise differences. For linear and logistic regression, the authors provide theoretic guarantees of NCM . Empirically, synthetic experiments validate the theory’s trade‑offs, and linear probing on CLIP features shows gains on ColoredMNIST, Waterbirds‑CF, and PACS.

**Strengths:**

- The proposed method, Noisy Counterfactual Matching (NCM), is well-motivated and intuitive—it estimates a “spurious” subspace from pairwise differences and removes it via projection.
- NCM is sample efficient. This is often easier to obtain in practice.
- The implementation of NCM is simple, and has potential to be applied to different applications.
- The method addresses the biggest practical challenge—noisy counterfactual pairs—and remains robust through a simple truncated-SVD design with a provable error bound.
- The theoretical analysis is solid in the linear setting, and synthetic results nicely match the theory.
- Beyond the linear case, real-data experiments show consistent gains for CLIP linear probes, suggesting the approach has broader practical potential beyond its theoretical contribution.

**Weaknesses:**

- The projection removes only observed linear directions of domain shift, leaving potential nonlinear or unseen correlations unaddressed.

**Questions:**

None.

---

> ### Author Response · Authors · 2025-11-21
> **Thank you for your valuable feedback**
>
> We appreciate the positive feedback and the recognition of our work. Regarding the linearity assumption: if the relationship between $X$ and $Y$ is nearly linear, nonlinearity can be characterized by the noise term. We acknowledge that for general nonlinear regimes, the denoising process is less direct, and we do not claim our current theory fully covers that case.
>
> However, our analysis suggests that a spectrum-aware, sparsity-inducing treatment of noisy pair differences (akin to $\ell_1$-type thresholding on singular values) is more appropriate than a simple $\ell_2$ matching loss. This is conceptually distinct from MatchDG-style $\ell_2$ penalties, as our method explicitly exploits the low-rank spurious structure and its spectral gap. Furthermore, we argue that the low-rank spurious signal should be estimated  from the difference between invariant pairs, not the predictivity of the features. Otherwise, the model risks using spurious correlations if it is anti-causal and predictive. This finding also demonstrates that simply reducing the dimension of the latent representation is insufficient.

---

### Meta-Review · Area_Chair_k3uN · 2025-12-29

**Summary:**

The authors propose Noisy Counterfactual Matching (NCM) for domain generalization, a data-centric approach which enforces prediction invariance across noisy counterfactual/invariant data pairs.

The Reviews think that the ideas of invariant data pairs for learning algorithmic approaches are appealing, theoretical finding results w.r.t. linear models are appreciated. However, the critical concerns on counterfactual/invariant data pairs should be addressed rigorously, and should be placed at the center of the work, which the proposal is entirely built upon. Additionally, it is better to distinguish its contributions with related works. Overall, we think the submission falls short to the bar. The authors may consider comments from the Reviewers to improve the submission.

**Reviewer Concerns:**

The Reviewers have some following concerns:

+ Reviewer PwUU: unaddressed potential nonlinear/unseen correlations beyond observed linear projections.

+ Reviewer zQ3E: the core idea, i.e., enforcing invariance by matching, orthogonality constraints, is incremental; weak support from theoretical finding results relying on linear models with idealized “counterfactual pairs”; strong assumptions, i.e., access to approximate counterfactual pairs or noisy invariant pairs; weak support from empirical evidence; lack analysis on orthogonality constraint extension.

+ Reviewer vnFv: novelty; weak experimental setting; access to counterfactual pairs

+ Reviewer FWNp: theoretical finding results limited to linear structural causal models; strong assumption on invariant data pair collection/ prior knowledge of all possible spurious features; presentation; counterfactual pairs v.s. invariant pairs; dependence of invariant data pairs on the optimal hypothesis; relation to augmented data for alignment; linear assumption restriction for theoretical finding results;

**Reviewer Scores:**

The authors clarify its novelty comparing to IRM and MatchDG; linear setting; availability of invariant pairs; experimental setting with linear model. Overall, we think the authors may partially address certain concerns about the linear setting; however, the impact of theoretical finding results are limited to linear structural causal models. It is better to emphasize the contributions by distinguishing from related works rigorously. The critical concerns on the invariant data pairs are important, which should be placed at the central of the work.

---

### Decision · Program_Chairs · 2026-01-26

Reject